# Transmembrane coupling of liquid-like protein condensates

Yohan Lee [1,5], Sujin Park [2,5], Feng Yuan [1], Carl C. Hayden[1], Liping Wang[3], Eileen M. Lafer[3], Siyoung Q. Choi[2] & Jeanne C. Stachowiak [1,4]

Liquid-liquid phase separation of proteins occurs on both surfaces of cellular membranes during diverse physiological processes. In vitro reconstitution could provide insight into the mechanisms underlying these events. However, most existing reconstitution techniques provide access to only one membrane surface, making it difficult to probe transmembrane phenomena. To study protein phase separation simultaneously on both membrane surfaces, we developed an array of freestanding planar lipid membranes. Interestingly, we observed that liquid-like protein condensates on one side of the membrane colocalized with those on the other side, resulting in transmembrane coupling. Our results, based on lipid probe partitioning and mobility of lipids, suggest that protein condensates locally reorganize membrane lipids, a process which could be explained by multiple effects. These findings suggest a mechanism by which signals originating on one side of a biological membrane, triggered by protein phase separation, can be transferred to the opposite side.

The spontaneous assembly of proteins into liquid-like condensates plays an important role in numerous biological processes, from chromosomal organization and regulation of gene expression to protein synthesis[1–4]. This phenomenon, which was initially observed for proteins residing in the cellular cytosol or in the nucleus, is now known to play a role in the assembly of diverse membrane-bound structures including the immunological synapse[5–7], focal adhesions[8], cell-cell junctions[9,10], and endocytic vesicles[11]. In each of these examples, proteins on both surfaces of the lipid bilayer are thought to play a role in the condensate-driven assembly of biological structures. For example, during assembly of the immunological synapse, phase separation of linker for activation of T cells (LAT) on the cytoplasmic side of the membrane is likely to be reinforced by interactions among the extracellular domains of receptors[12]. Similarly, when focal adhesions are established, polymerization of integrin domains on the extracellular surface is complemented by phase separation of cytoplasmic proteins, such as talin, kindlin, and vinculin, on the intracellular side of the membrane[8]. Finally, during the initiation of endocytic vesicles, phase separation of early endocytic proteins is likely complemented by

condensation of transmembrane cargo proteins, such as receptor tyrosine kinases, several of which have recently been found to form liquid-like assemblies on membrane surfaces[13]. Collectively, these examples suggest that proteins frequently condense on both surfaces of lipid membranes to control diverse membrane-associated biological events. However, in vitro reconstitution of such processes has been challenging when using conventional model membrane systems, such as lipid vesicles[14,15] and supported lipid bilayers[5–7,16–18], each of which provides access to only one side of the membrane[14–18]. Therefore, the development of a new system to study protein phase separation on both sides of the membrane at the same time is needed.

In this work, we present suspended planar lipid membrane arrays on the surfaces of electron microscopy grids and use them to observe protein phase separation on both sides of the membrane surface. Interestingly, we observe that phase separation of model proteins is highly coupled across the membrane surface, such that protein-enriched regions on one side of the membrane tightly colocalize with protein-enriched regions on the opposite side of the membrane. Throughout the paper, we will refer to this transmembrane

[1]Department of Biomedical Engineering, The University of Texas at Austin, Austin, TX, USA. [2]Department of Chemical and Biomolecular Engineering, Korea Advanced Institute of Science and Technology (KAIST), Daejeon, Republic of Korea. [3]Department of Biochemistry and Structural Biology, The University of Texas Health Science Center at San Antonio, San Antonio, TX, USA. [4]Department of Chemical Engineering, The University of Texas at Austin, Austin, TX, USA. [5]These authors contributed equally: Yohan Lee, Sujin Park. ✉e-mail: jcstach@austin.utexas.edu

colocalization of protein-rich domains as transmembrane coupling. These findings suggest a fundamental mechanism of information transfer across lipid bilayer, which may play a role in initiating and stabilizing diverse protein complexes that assemble at membrane surfaces.

## Results

### The RGG domain of LAF-1 forms liquid-like condensates on freestanding planar membranes

Freestanding planar lipid membranes were created within the hexagonal holes of transmission electron microscopy (TEM) grids (Supplementary Fig. 1), as described previously[19]. Each grid contained 150 holes, each of which had a diameter of approximately 100 μm, enabling visualization of multiple independent membrane surfaces per field of view using fluorescence confocal microscopy. To investigate protein phase separation on planar lipid membranes, the RGG domain of the LAF-1 protein (RGG) was selected, as its participation in protein phase separation is well-established. We added N-terminal histidine-tagged RGG (his-RGG), labeled with Atto 488, at a concentration of 1 μM to freestanding planar membranes containing DGS-Ni-NTA lipids (5–25 mol%). Here, binding of his-RGG to the membrane was achieved through interactions between histidine and Ni-NTA (Fig. 1a). In our initial experiments, we sought to examine protein phase separation on only one surface of the bilayer. Therefore, we placed the TEM grid directly against a glass coverslip so that proteins only bound significantly to the top surface of the membrane. Importantly, RGG is highly soluble in aqueous buffers, such that phase separation in solution only occurs at concentrations greater than 10 μM[20,21], substantially above the concentration used in our experiments with membranes. In this way, we could clearly differentiate protein phase separation on membranes from phase separation in the surrounding solution, which was not observed under our experimental conditions.

Initially, the fluorescence intensity in the protein channel was homogeneous over the surface of the membrane, indicating uniform protein binding. However, after a few minutes, heterogeneity in the intensity of the protein channel appeared, such that brighter and dimmer regions began to coexist, indicating phase separation of the membrane-bound protein layer into protein-rich (brighter) and protein-poor (dimmer) phases. We also observed fusion and re-rounding upon contact between different protein-rich domains (Fig. 1b, c) and different protein-poor domains (Fig. 1d), leading to domain coarsening over time (Supplementary Movie. 1, 2). Interestingly, the protein-rich and protein-poor phases displayed a bicontinuous morphology in the early moments after protein binding (Fig. 1b–d, 2–3 min), suggesting that phase separation occurred through spinodal decomposition, rather than a nucleation and growth process[22–24]. In addition, protein-rich regions exhibited fluorescence recovery within a few seconds after photobleaching, with $t_{1/2}$ of 10.5 s and a mobile fraction of 79% (Fig. 1f). Collectively, these results suggest that two-dimensional, membrane-bound protein condensates of the RGG domain have liquid-like properties. Notably, because the membranes were composed of unsaturated lipids, DOPC, and DGS-Ni-NTA, with melting temperatures far below room temperature, phase separation of the lipid membrane is unlikely to play a role in this process[25]. However, the impact of protein phase separation on the organization of the lipids is explored later in this report.

### The strength of protein-protein interactions and the density of protein-membrane interactions determine the area fraction of the protein-rich phase

We next probed the sensitivity of membrane-bound protein condensates to changes in the strength of protein-protein interactions and the density of protein-membrane interactions. To vary the strength of protein-protein interactions, we changed the ionic strength of the buffer by controlling the sodium chloride (NaCl) concentration.

Specifically, increasing the salt concentration screens residue-residue interactions, hindering phase separation of RGG[20]. To vary the density of protein-membrane interactions, we changed the concentration of DGS-Ni-NTA lipids within the membrane.

Comparing the membranes within the grid upon the addition of his-RGG, we observed four distinct cases, with the later cases becoming more common as protein concentration increased and ionic strength decreased (Fig. 1g). In the first case, the grid hole was covered entirely by the protein-depleted phase. In the second case, the continuous phase was protein-depleted, while the dispersed phase was protein-enriched. In the third case, the continuous phase was protein-enriched, while the dispersed phase was protein-depleted. Finally, in the fourth case, the entire hole was covered by the protein-enriched phase. To characterize the impact of protein-protein and protein-lipid interactions on phase separation of membrane-bound proteins, we quantified the fraction of the grid holes that belonged to each case 15 min after protein addition. When phase separation was weakened by high ionic strength and low Ni-NTA concentration, the majority of grid holes belonged to cases 1 and 2. In contrast, when phase separation was strengthened by low ionic strength and high Ni-NTA concentration, the majority of the grid holes belonged to cases 3 and 4 (Fig. 1h). Similarly, the fraction of the membrane area covered by the protein-rich phase increased with decreasing NaCl concentration and increasing Ni-NTA concentration (Supplementary Fig. 2). The observation of multiple cases for the same condition did not arise from variation in lipid composition between the grid holes, as a non-phase separating protein, green fluorescent protein (GFP) bound each hole approximately equally (Supplementary Fig. 3). However, the variation could arise from imperfect mixing upon addition of protein to solution, leading to variation in the extent of protein binding across the grid. Taken together, these results demonstrate that the same variables that control phase separation in solution – local protein concentration and the strength of protein-protein interaction – also govern protein phase separation on membrane surfaces.

### Increasing salt concentration lowers the transition temperature for phase separation of RGG on membrane surfaces

A characteristic of systems that undergo phase separation is that the relative concentrations of macromolecules in the dilute and enriched phases become more similar to one another as the temperature of observation approaches the transition temperature[26]. To determine whether the membrane-bound condensates of the RGG domain display this behavior, we observed our system while increasing the temperature. At each temperature, the relative fluorescence intensities of the protein-enriched and protein-depleted phases provide a rough estimate of the relative protein concentration within each phase[11]. In particular, by using these concentrations to represent the ends of a tie line, an approximate temperature-concentration phase diagram can be constructed (Fig. 2c), where $C_{rich}$ is proportional to the protein concentration in the protein-enriched phase, and $C_{poor}$ is proportional to the protein concentration within the protein-depleted phase. We used this approach to map the phase diagram for membrane-bound condensates of RGG at an NaCl concentration of 100 mM. Here, the difference in intensity between the protein-enriched and protein-depleted regions was gradually lost as the temperature was raised from room temperature to 39 °C, the highest temperature we could achieve in our microscopy system. At 39 °C, some protein-enriched condensates of low contrast remained, suggesting that the transition temperature was greater than 39 °C at this NaCl concentration (Fig. 2a). In contrast, a complete dissolution of the protein-enriched phase was observed at 32 °C with an NaCl concentration of 200 mM (Fig. 2b). Our observation that the transition temperature decreases with increasing NaCl concentration is consistent with the ability of NaCl to screen residue-residue interactions between RGG proteins, hindering protein condensation[20].

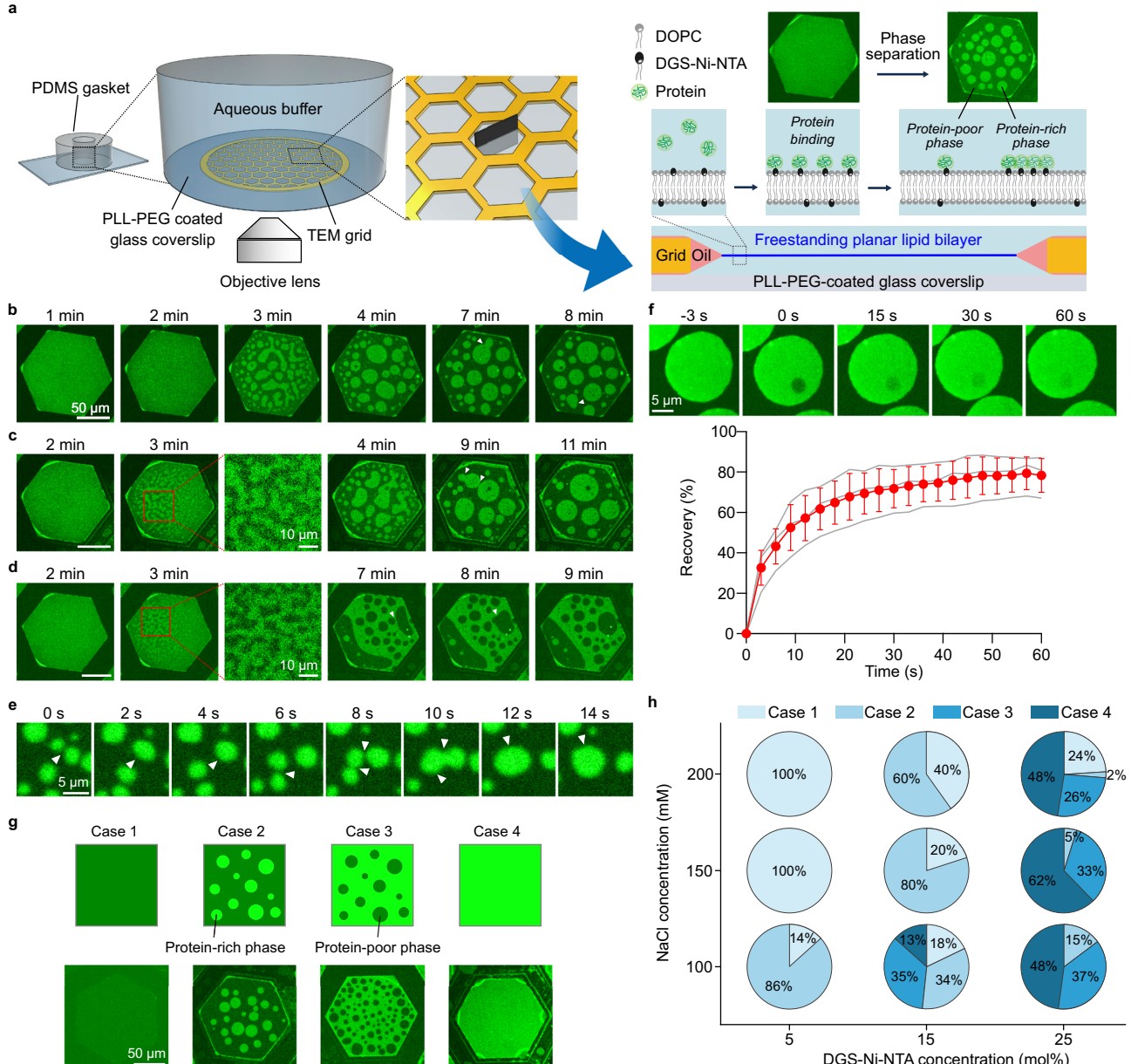

**Fig. 1 | Phase separation of RGG domains on freestanding planar membranes results in liquid-like protein assemblies. a** Left: Schematic of the freestanding planar membrane array system. Inside a PDMS chamber, a transmission electron microscopy (TEM) grid with hexagonal holes is immersed in an aqueous buffer, and freestanding planar membranes are created within multiple hexagonal holes. Right: Cross-section of the freestanding membrane spanning a single hexagonal hole. Once proteins (histidine-tagged) are added to the aqueous buffer in the chamber, they bind to the membrane through histidine-nickel interactions and phase separate into protein-rich and protein-poor phases. **b–d** Representative images showing spinodal decomposition and the increase in domain size. The elapsed time since protein addition is indicated above each column. Scale bars, 50 μm. **e** Fusion events between different protein-rich domains on the membrane over time. Scale bar, 5 μm. White arrowheads indicate fusion events (**b–e**). **f** Top: Representative images

showing fluorescence recovery after photobleaching (FRAP) of a protein-rich domain on the membrane. Bottom: Corresponding FRAP profile. Data are presented as mean values ± SD ($n = 3$). Gray lines represent each independent recovery curve. Scale bar, 5 μm. **g** Top: Schematic of four possible cases, from case 1 to 4, when phase-separating proteins are associated with membranes. Bottom: Representative images for each case. Scale bar, 50 μm. **h** Percentage of individual cases as a function of DGS-Ni-NTA concentration in the membrane and NaCl concentration in the aqueous buffer. More than 130 lipid membranes were analyzed from three independent experiments for each condition. Membrane composition: 75 mol% DOPC, 25 mol% DGS-Ni-NTA (**b–d**), 85 mol% DOPC, 15 mol% DGS-Ni-NTA (**e, f**). 0.5 mol% Texas Red-DHPE was added for all. Buffer: 25 mM HEPES, 100 mM NaCl, pH 7.4. 1 μM of his-RGG labeled with Atto 488 was used. Source data are provided as a Source Data file.

## Transbilayer coupling of protein condensates occurs when proteins phase separate on both sides of the membrane at the same time

So far, we have only allowed RGG proteins to bind and phase separate on one surface of the membrane. What will happen if phase separation occurs on both surfaces of the lipid bilayer at the same time? By introducing thin spacers between the TEM grid and the glass coverslip

at the bottom of our imaging chamber, both sides of the membrane were exposed to proteins dissolved in the surrounding aqueous medium (Fig. 3a). Within a few minutes after proteins bound to both sides of the membrane, we began to observe heterogeneity in the distribution of proteins over the membrane surface. Protein-enriched and protein-depleted regions appeared, as before, with a key difference. When proteins bound to only one side of the membrane, we

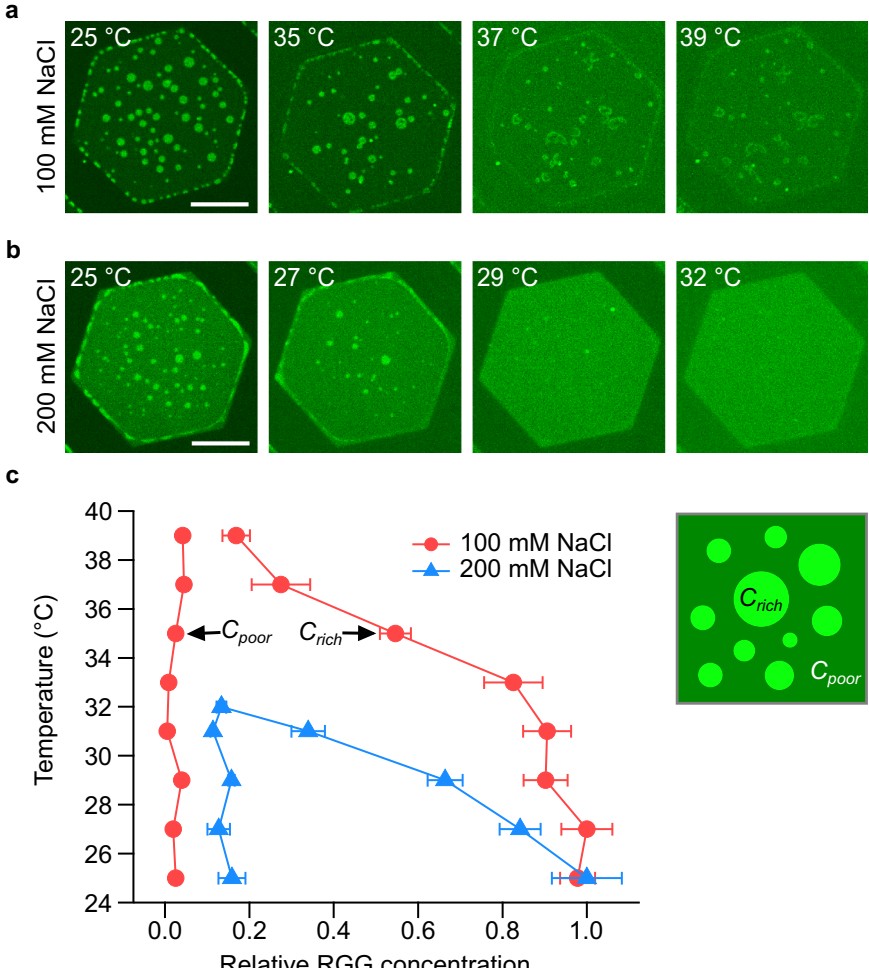

**Fig. 2 | Concentration-temperature phase diagram with varying NaCl concentrations. a, b** Representative images at different temperatures, indicated at the top left for each image, after adding 1 μM of his-RGG, labeled with Atto 488, with the NaCl concentration of 100 mM (**a**) and 200 mM (**b**). Membrane composition: 85 mol% DOPC, 15 mol% DGS-Ni-NTA, and 0.5 mol% Texas Red-DHPE. Scale bars, 50 μm. **c** Phase diagram of RGG condensates on the membrane with varying temperatures depending on NaCl concentration. $C_{rich}$ and $C_{poor}$ represent the concentration of protein-rich and protein-poor phases, respectively. Data are presented as mean values ± SD from analyzing 3–10 protein-rich and protein-poor regions from three independent experiments at each temperature. Source data are provided as a Source Data file.

observed two levels of protein intensity, the brighter of which corresponded to the protein-enriched phase, while the dimmer corresponded to the protein-depleted phase (Fig. 1). In contrast, when proteins bound to both sides of the membrane, there were three levels of intensity, which we will denote as the brightest, medium-bright, and dimmest regions (Fig. 3b). Interestingly, we observed that the intensity difference between the dimmest and medium-bright regions was similar to that between medium-bright and the brightest regions. The simplest explanation for these observations is that the regions of dimmest intensity represent areas where proteins on both membrane surfaces are in the protein-depleted phase. In contrast, the regions of brightest intensity represent areas where proteins on both membrane surfaces are in the protein-enriched phase. Finally, the regions of medium intensity represent areas where proteins on one side of the membrane are in the protein-depleted phase, and those on the other side of the membrane are in the protein-enriched phase (Fig. 3c). Note that this observation can also be described as the coexistence of three distinct phases[27]. Importantly, the spacer used to support the suspended bilayer was thin, such that the bilayer remained within the limited working distance of the high magnification microscope objective. For this reason, protein binding to the bottom surface was slower, likely leading to smaller protein-enriched regions on the bottom surface compared to the top surface, particularly at early times (Fig. 3c cartoon). Interestingly, regions of the brightest intensity were typically surrounded by regions of medium intensity. This observation suggests that protein phase separation on one side of the membrane was coupled to protein phase separation on the other side of the membrane, such that phase-separated regions tended to colocalize across the membrane boundary.

If protein phase separation is coupled across the membrane bilayer, what will happen when two protein-enriched domains cross paths? If they are on the same side of the membrane, they should fuse upon contact. In line with this prediction, Fig. 3d shows two medium-bright domains that each contain one or two domains of the brightest intensity. When the medium-bright domains contacted one another, they fused within seconds, similar to our observation of fusion between domains on a single side of the membrane (Fig. 1b–e). This fusion event brought the domains of the brightest intensity into contact with one another, also resulting in their fusion (Fig. 3d, 90 s, Supplementary Movie 3). These observations demonstrate that protein-rich domains on the same side of the membrane fuse upon contact (Fig. 3e), which provides further evidence that protein phase separation is occurring on both surfaces of the membrane.

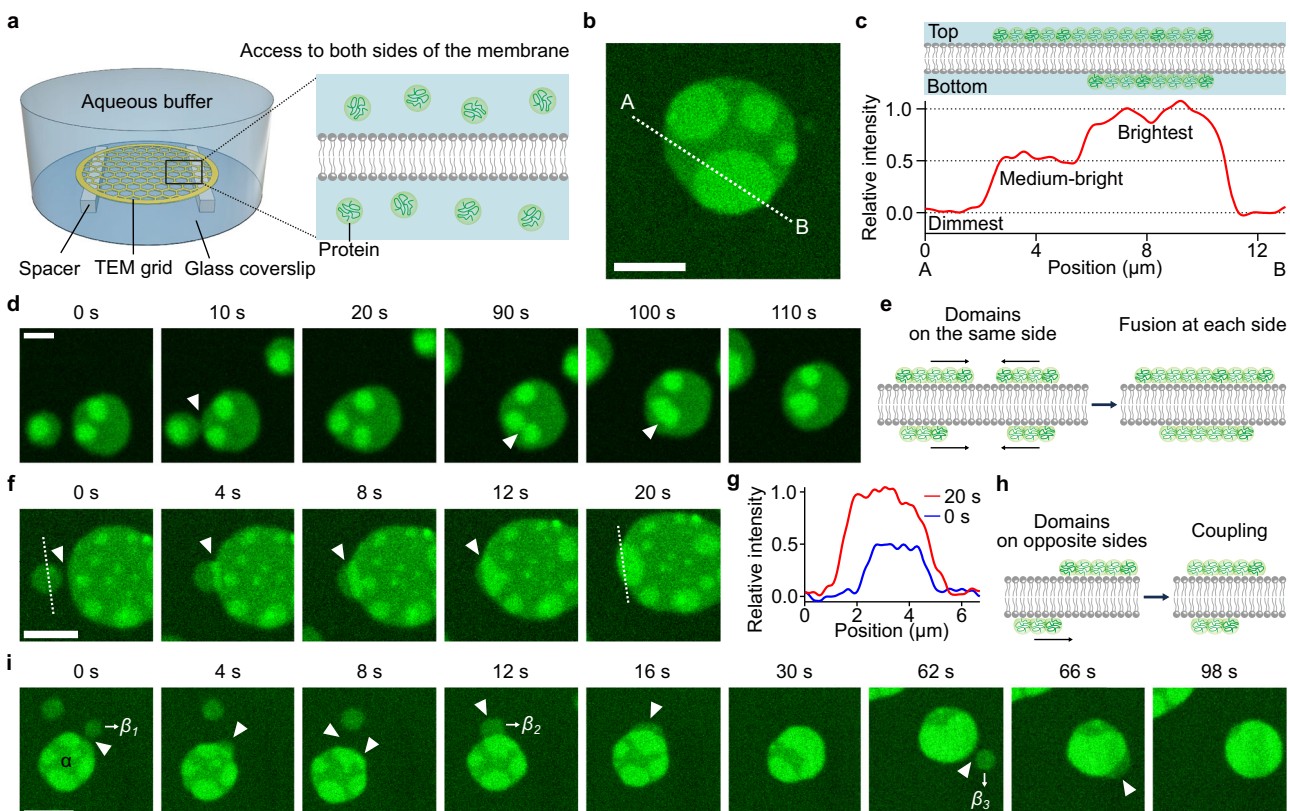

**Fig. 3 | Protein phase separation on both membrane surfaces leads to transmembrane coupling of protein-rich domains. a** Schematic of the system where protein access to both sides of the membrane is possible by placing spacers between the TEM grid and the glass coverslip. **b** Representative microscopic images of protein regions with three different brightnesses: dimmest, medium-bright, and brightest regions. **c** Cartoon of cross-section of the membrane (Top) and relative intensity profile (Bottom) along the dotted line (from A to B) in **b**, where regions with relative intensity of 0, 0.5, and 1.0 correspond to the dimmest, medium-bright, and brightest regions, respectively. Relative intensity ($I_R$) was defined as $I_R = (I - I_D)/(I_B - I_D)$, where I, $I_D$, and $I_B$ indicate the fluorescence intensity of the region of interest, the intensity of the dimmest region, and the intensity of the brightest region, respectively. **d–i** Representative microscopic images over time and cartoons showing dynamic changes when different protein domains were spatially overlapping. Fusion occurred when domains were on the same side of the membrane (**d, e**), and coupling occurred when domains were on opposite sides of the membrane (**f–i**). **g** Relative intensity profile along the dotted white lines in the first (0 s, blue) and the last (20 s, red) images in **f**. White arrowheads indicate fusion (**d**) or coupling (**f, i**) spots. 1 μM of his-RGG labeled with Atto 488 was used. Buffer: 25 mM HEPES, 100 mM NaCl, pH 7.4. Membrane composition: 85 mol% DOPC, 15 mol% DGS-Ni-NTA, and 0.5 mol% Texas Red-DHPE. Scale bars, 5 μm. Source data are provided as a Source Data file.

Next, if coupling is occurring, when two protein-enriched domains are on opposite sides of the membrane and cross paths, they should become coupled. In line with this prediction, Fig. 3f (Supplementary Movie 4) shows a medium-bright region that meets another, larger medium-bright region, which also includes several smaller regions of the brightest intensity. As the two medium-bright regions meet, rather than fusing, a stable overlap region develops, which has a brightness similar to the brightest regions within the same image, showing a stepwise increase in relative intensity from 0.5 to 1.0 (Fig. 3g). Based on our previous observations, if these two domains were on the same side of the membrane, then fusion between them would have occurred with the brightness unchanged and the overall domain size increasing. Instead, the observed increase in brightness, without a change in domain size, strongly suggests that medium-bright domains on opposite sides of the membrane crossed paths, recognized each other, and became coupled (Fig. 3h). Similarly, we observed consecutive domain coupling events, which eventually led to the full transmembrane coupling of protein-enriched domains (Fig. 3i, Supplementary Movie 5). Specifically, we initially observed a protein region, denoted as the α region, composed of both medium-bright domains and domains of brightest intensity. Then, between 0 s and 8 s, the medium-bright part of the α region and another smaller medium-bright region, denoted as β₁, crossed paths. Since brightness increased in the overlapped region without an increase in domain size, it

appeared that these two overlapping domains were on different sides of the membrane and became coupled. After that, two more similar coupling events were observed (12–16 s and 62–66 s), as the β₂ and β₃ domains, initially having medium brightness, became coupled, such that full coupling was achieved, resulting in a single domain of brightest intensity (Fig. 3i). Notably, we observed more domain coupling over time, which likely represents the approach of the system to equilibrium.

In all cases, protein-enriched domains either merged, if on the same side of the membrane, or became stably coupled, if on opposite sides of the membrane. In contrast, if protein-enriched regions on opposite sides of the membrane were uncoupled, we would expect them to diffuse over one another without becoming coupled. Such an event would transiently create a region of the brightest intensity during the time that the domains passed over one another on opposite sides of the membrane but would not lead to stable coupling. Events with these characteristics were never observed in our experiments. Collectively, our observations demonstrate that protein-enriched regions on the surfaces of suspended lipid bilayers undergo transmembrane coupling, such that protein phase separation on one side of the membrane frequently colocalizes with protein phase separation on the opposite side of the membrane.

How can protein condensates on different sides of the membrane recognize each other and become coupled? Because the proteins

attach peripherally to the membrane surface via the histidine-Ni-NTA interaction, there is no direct contact between proteins on the two sides of the membrane, suggesting that coupling occurs indirectly through protein-lipid interactions. Therefore, we next examined the impact of protein condensates on the behavior of the membrane lipids.

### Lipid probes are depleted from protein-rich regions

To probe the impact of protein phase separation on the lipids, we examined the distribution of a fluorescent lipid probe, Texas Red-DHPE, which we included at 0.5 mol% in the solvent mixture used to create the suspended membranes. Interestingly, the intensity distribution in the lipid channel was opposite of that in the protein channel, such that the brightest regions in the protein channel, which are the coupled protein-enriched regions, corresponded to the dimmest regions in the lipid channel. Likewise, the dimmest regions in the

protein channel, which are the protein-depleted regions, corresponded to the brightest regions in the lipid channel (Fig. 4a). These observations suggest that protein phase separation results in the depletion of the probe lipids from the underlying membrane. In particular, in the lipid probe channel, the increase in fluorescence intensity from the dimmest to medium-bright regions was comparable to the increase from the medium-bright to the brightest regions (Fig. 4b). This comparison suggests that protein condensation on both sides of the membrane resulted in about twice as much depletion of the probe lipid as protein condensation on one side of the membrane.

To check if a small amount of residual oil (hexadecane) in our membrane had an effect on lipid probe exclusion, we replaced hexadecane ($C_{16}H_{34}$) with squalane ($C_{30}H_{62}$), which has a longer hydrocarbon chain that has been shown to greatly reduce the amount of trapped oil in the bilayer, leading to membranes with a very low solvent content[28–30]. We observed a very similar depletion of the probe

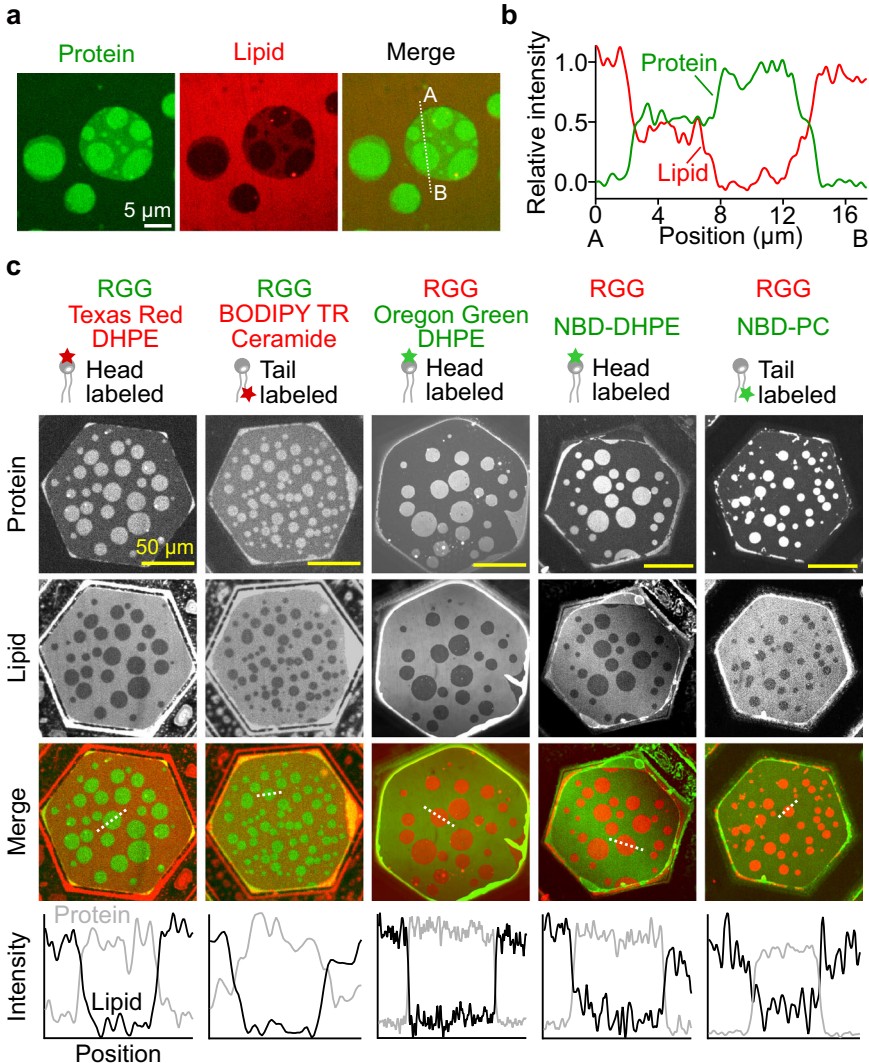

**Fig. 4 | Multiple lipid probes partition away from protein-rich regions.**
**a** Representative microscopic images showing regions with three different brightness from protein and lipid channels. 1 μM of his-RGG labeled with Atto 488 was applied to both sides of the membrane. Membrane composition: 85 mol% DOPC, 15 mol% DGS-Ni-NTA, and 0.5 mol% Texas Red-DHPE. Scale bar, 5 μm.
**b** Relative intensity profile along the dotted line (from A to B) in the merged channel in **a**, where green and red lines indicate relative intensity from the protein and lipid channels, respectively. See the caption of Fig. 3c for the definition of relative intensity. **c** Representative microscopic images after the addition of 1 μM of his-RGG

to the top side only of each membrane containing either head labeled (Texas Red-DHPE, Oregon Green-DHPE, and NBD-DHPE) or tail labeled (BODIPY TR-Ceramide and NBD-PC) lipid probes. His-RGG was labeled with Atto 488 or Atto 594, depending on the lipid probes used. The relative fluorescence intensity profile along the dotted line in each merged channel image is shown at the bottom, where gray and black lines represent the intensity from protein and lipid channels, respectively. Buffer: 25 mM HEPES, 100 mM NaCl, pH 7.4. Membrane composition: 85 mol% DOPC, 15 mol% DGS-Ni-NTA, and 0.5–1.0 mol% lipid probe. Scale bars, 50 μm. Source data are provided as a Source Data file.

lipid from protein-enriched regions using squalane (Supplementary Fig. 4), suggesting that the depletion of the probe lipid cannot be explained by the inclusion of oil in the bilayer.

Next, we tested the generality of lipid probe exclusion from protein-enriched phases by examining the partitioning of several additional probe lipids in membranes on which protein phase separation was taking place. Texas Red-DHPE emits in the orange/red region of the spectrum and is covalently conjugated to the head group of a phospholipid. In addition to this probe, we also examined Oregon Green-DHPE and NBD-DHPE, which are similar to Texas Red-DHPE in that the fluorophore is conjugated to the phospholipid head group, except that the incorporated fluorophores emit in the green region of the spectrum. We also examined BODIPY TR-Ceramide (orange/red) and NBD-PC (green), which incorporate fluorophores conjugated to the lipid tail group, eliminating direct interactions between proteins and fluorescent probes. We examined the partitioning of each of these probe lipids in membranes on which phase separation of RGG was taking place on one side of the membrane. In each case, we observed that the dimmer regions in the lipid channel corresponded to protein-enriched regions that were brighter in the protein channel (Fig. 4c). These data illustrate that a diverse set of probe lipids are depleted from regions of the membrane on which protein-enriched phases exist. We also confirmed that lipid probe partitioning was observed when unlabeled his-RGG was used (Supplementary Fig. 5), demonstrating that the reduction in probe lipid intensity in protein-rich regions could not be explained by spectral interactions between their respective fluorophores.

Why are diverse probe lipids excluded from the protein-enriched phase? Probe exclusion arises due to the local reorganization of lipids, which could encompass changes in lipid packing[31,32], lipid composition (enrichment of DGS-Ni-NTA lipids in protein-rich regions), or the degree of lipid hydration owing to the presence of proteins[33], each via protein-lipid interactions. To quantify the extent of probe lipid exclusion, we defined a partition coefficient ($K_P$) as $K_P = I_B/I_D$, where $I_B$ and $I_D$ indicate the fluorescence intensity of the brighter and the dimmer regions in the lipid channel after subtracting the background intensity, respectively[34]. Based on the definition of $K_P$, the more the probes are excluded from the dimmer region, the higher the $K_P$ value will become.

Interestingly, the extent of probe lipid (Texas Red-DHPE) exclusion remained constant across a range of DGS-Ni-NTA concentrations (5, 15, 25 mol%) (Supplementary Figs. 6 and 7). Meanwhile, the area fraction of protein-rich regions increased over the same range of DGS-Ni-NTA concentrations (Supplementary Fig. 2). One possible explanation for these two findings is that the protein-rich region becomes enriched to some degree in DGS-Ni-NTA lipids. Once a stable degree of enrichment is achieved within these regions, increasing the DGS-Ni-NTA concentration increases the area fraction of the protein-rich phase. However, our data at present cannot distinguish between several possible mechanisms for probe lipid exclusion, listed above. Note that in our past work with GUVs, the head-labeled probe lipid, Texas Red-DHPE, was enriched rather than depleted in protein-rich regions of the membrane[15]. While the reasons for this difference remain unclear, it is possible that the higher membrane tension in suspended membranes, relative to GUVs[35], permits greater shifts in lipid packing upon protein condensation.

To further assess whether DGS-Ni-NTA lipids were required for the observed probe partitioning, we examined an alternative means of recruiting the RGG protein to membrane surfaces. Specifically, we created a recombinant protein chimera that linked the epsin1 N-terminal homology (ENTH) domain and RGG, ENTH-RGG. The ENTH domain is known to bind phosphatidylinositol-4,5-bisphosphate (PI(4,5)P2) lipids[36]. Therefore, to recruit ENTH-RGG to the membrane surface, we incorporated PI(4,5)P2 lipids into the membrane. Here, the exclusion of lipid probes (BODIPY TR Ceramide) from protein-rich regions was observed, with a somewhat smaller partition coefficient ($K_P$) of 1.24 compared to the $K_P$ value of 1.60 for the same probe when DGS-Ni-NTA was used for recruitment of his-RGG. Transmembrane coupling of ENTH-RGG condensates was also observed (Supplementary Fig. 8). These results suggest that the exclusion of probe lipids from the protein-rich phase, as well as transmembrane coupling of protein condensates, occur regardless of the specific binding interaction used to recruit proteins to the membrane surface. However, the observed variation in the extent of probe lipid exclusion suggests that protein-lipid binding interactions play a role in transbilayer coupling.

## Transmembrane coupling of protein condensates correlates with changes in lipid organization and mobility

To further investigate the effect of protein condensates on lipid organization, we compared the dynamics of condensate fusion for uncoupled (one-sided) and coupled (two-sided) condensates. (Fig. 5b, c). Specifically, by imaging the lipid probe channel during fusion events, we quantified the relaxation after fusion by measuring the aspect ratio of two coalescing domains over time, which followed an exponential decay[37]. We observed that coupled regions had a longer characteristic relaxation time than uncoupled ones (Fig. 5d), implying that protein condensates reduced the fluidity of the resulting protein-membrane composite.

If protein condensation reduces the fluidity of the membrane, then the lipids beneath the condensates should experience slower diffusion in comparison to lipids in regions with more dilute protein binding. To test this prediction, we used fluorescence recovery after photobleaching (FRAP) to examine the diffusion of probe lipids within: (i) protein-depleted regions (brightest in the probe lipid channel), (ii) regions with uncoupled protein-enriched domain on a single side of the bilayer (medium-bright regions in the probe lipid channel), and (iii) regions with coupled protein-enriched domains on both sides of the bilayer (dimmest regions in the probe lipid channel) (Fig. 5e). For these experiments we used the NBD-PC probe lipid, which is tail-labeled, such that the NBD fluorophore is unlikely to interact with the protein layer. Unlabeled RGG proteins were used to eliminate the possibility of spectral cross-talk with the probe lipid. The time required for 50% recovery after bleaching, $t_{1/2}$, was obtained from the recovery curve for each region (Fig. 5f). We observed that the brightest lipid region exhibited $t_{1/2}$ of 0.48 s, similar to that of protein-free control membranes (0.43 s), whereas medium-bright and dimmest lipid regions displayed longer $t_{1/2}$ of 0.75 s and 1.03 s, respectively (Fig. 5g). Slower recovery suggests that lipid mobility within the region of interest was reduced. From these results, we conclude that the diffusivity of the probe lipid was greatest in the protein-depleted phase, decreased in regions of the membrane with protein-enriched phase on one side, and decreased further in regions of the membrane with coupled protein-enriched phases on both sides. The observed differences in lipid mobility could result from differences in lipid packing but could also arise from a combination of effects including differences in lipid composition, such as enrichment of DGS-Ni-NTA lipids in protein-rich regions, frictional interactions between proteins and lipids, and modified local bending rigidity or local spontaneous curvature due to membrane-bound proteins[38].

Next, we evaluated the impact of protein condensation on probe lipid partitioning by varying the strength of protein-protein interactions. Measurements of the partition coefficient ($K_P$) were conducted for the case in which protein phase separation took place on only one side of the membrane after applying 1 μM of his-RGG, where Texas Red-DHPE was used as a lipid probe. We observed that decreasing the concentration of NaCl, which is expected to strengthen interactions among RGG domains, resulted in increasing values of $K_P$ (Fig. 5h). This observation suggests that strengthening protein-protein interactions results in greater depletion of probe lipids from the underlying lipid bilayer. However, it is important to note that the change in

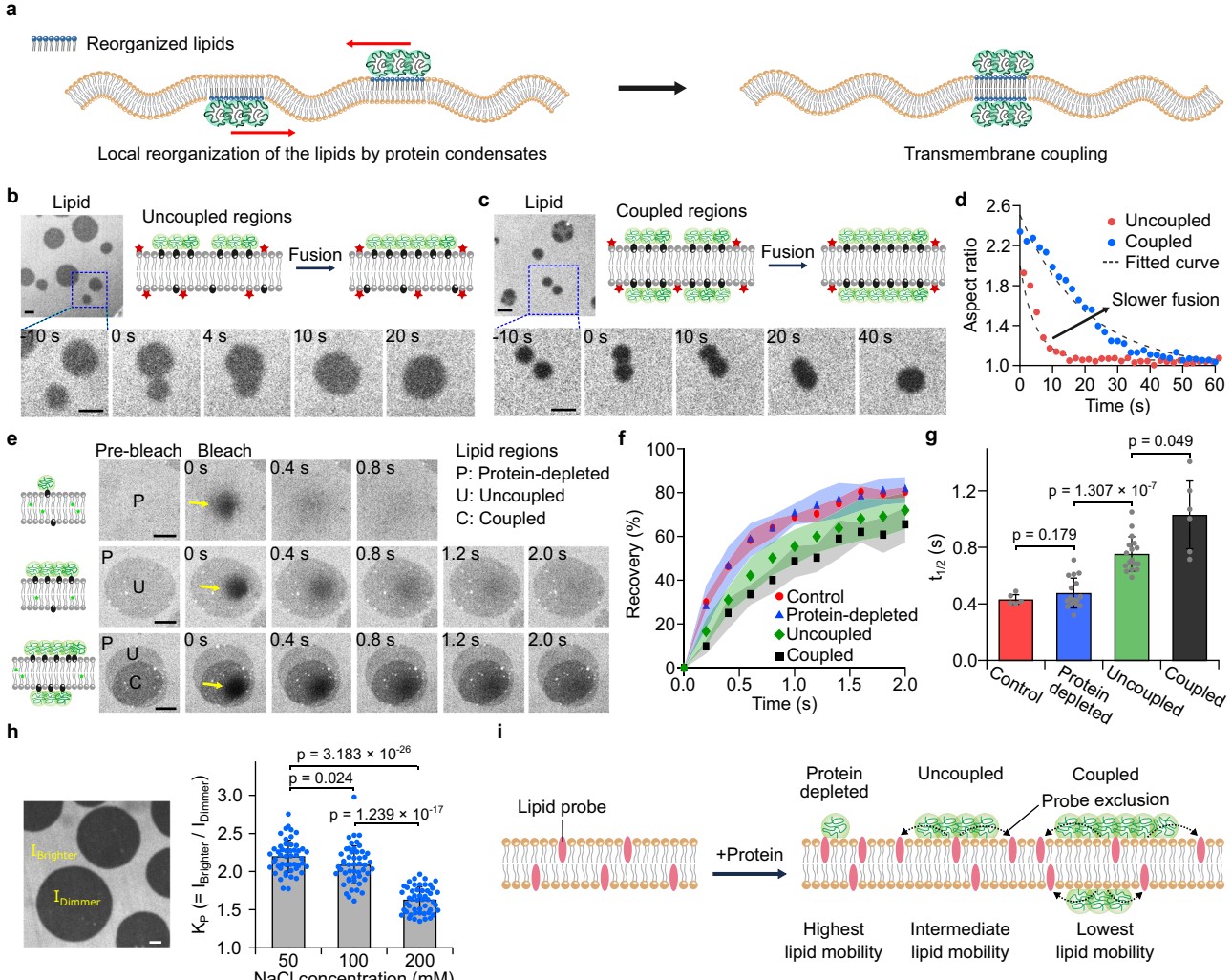

**Fig. 5 | Transmembrane coupling of protein condensates correlates with changes in lipid organization and mobility. a** Schematic of the transmembrane coupling process. Left: Protein condensates induce local reorganization of lipids (blue lipid heads). Right: Protein condensates on different sides of the membrane become coupled. Representative images from lipid channels and cartoons showing fusion events of two uncoupled regions (**b**) and two coupled regions (**c**) of membrane-protein composites over time. **d** The aspect ratio changes over time during relaxation after the fusion of two regions. Red circles indicate aspect ratio change for uncoupled regions and blue circles for coupled regions. The dotted lines represent an exponential fit: $y(t) = A + B*exp(-t/\tau)$. **e** Images in the lipid channel showing fluorescence recovery for protein-depleted, uncoupled, and coupled regions. Yellow arrows indicate photobleached regions. **f** FRAP profile for the control (protein-free membrane, red circles, $n = 5$), Protein-depleted (blue

triangles, $n = 16$), uncoupled (green diamonds, $n = 17$), and coupled (black squares, $n = 6$) regions. **g** Corresponding $t_{1/2}$, time required for 50% of fluorescence recovery, from FRAP profile in **f**. **h** Left: Lipid channel image showing both brighter and dimmer regions for calculating the partition coefficient of the lipid probe. Right: Partition coefficients ($K_P$) as a function of NaCl concentrations. ($n = 50$ from 3 independent experiments for each condition). Data are presented as mean values ± SD (**f**–**h**) **i** Schematic of our observation that protein condensates create lipid regions with reduced lipid mobility. Brackets in **g** and **h** show statistically significant comparisons using an unpaired, two-tailed Student's *t* test. Above each bracket are *p*-values. Membrane composition: 85 mol% DOPC, 15 mol% DGS-Ni-NTA with 0.5 mol% Texas Red-DHPE (**b**, **c**, **h**) or NBD-PC (**e**). 1 μM of unlabeled his-RGG was used. Scale bars, 5 μm. Source data are provided as a Source Data file.

NaCl concentration also impacts the area fraction of the protein-enriched phase (Supplementary Fig. 2), and may alter its lipid composition.

Collectively, our findings suggest that protein condensates reduce lipid mobility (Fig. 5i) and facilitate lipid probe partitioning by reorganizing membrane lipids. How might the local reorganization of lipids contribute to the transmembrane coupling of protein condensates? It is well known that lipid domains in opposing membrane leaflets frequently become coupled, or registered, by various mechanisms including line tension, membrane fluctuation, and spontaneous curvature[27,39–42]. Transmembrane coupling of protein condensates could, in principle, arise from lipid domain coupling, which could be triggered by the local reorganization of lipids. Given the diversity of factors that could contribute to our findings, future work

by the community, including theoretical studies[43,44], will be needed to isolate the specific mechanisms responsible for transmembrane coupling of protein condensates.

## Transmembrane coupling is a general phenomenon when proteins phase separate on membrane surfaces

To examine the generality of transmembrane coupling of protein condensates, we examined membrane-bound phase separation of an additional protein domain known to form liquid-like condensates, the low complexity domain of the fused in sarcoma (FUS LC) protein[45]. We observed liquid-like assemblies of FUS LC on membranes, which fused and re-rounded upon contact. Additionally, we observed a similar trend of probe lipid (Texas Red-DHPE) partitioning, where lipid probes were depleted from protein-enriched regions

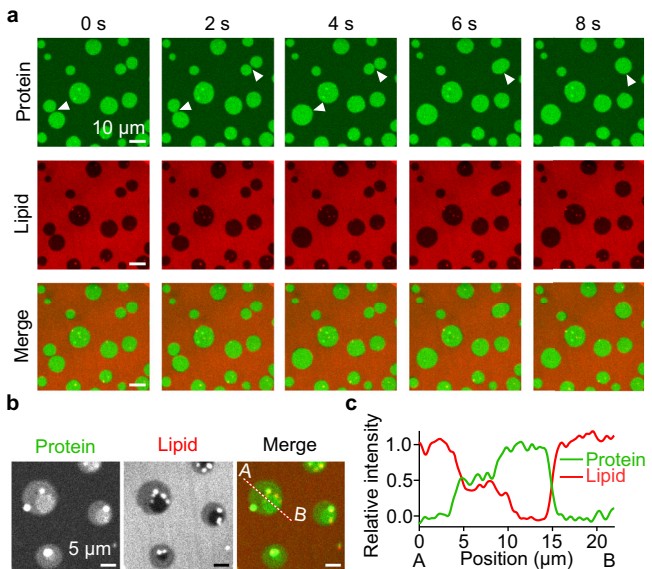

**Fig. 6 | Phase separation of FUS LC domains on membranes. a** Representative microscopic images showing the fusion of protein-rich regions. White arrowheads indicate fusion events. **b** Representative microscopic images showing transbilayer domain coupling. Scale bars, 5 μm. **c** Relative intensity profiles along the dotted lines (from A to B) in the merged channel in **b**, where green and red lines represent the intensity from the protein and lipid channels, respectively. See the caption of Fig. 3c for the definition of relative intensity. 500 nM of his-FUS LC, labeled with Atto 488, was used. Membrane composition: 80 mol% DOPC, 20 mol% DGS-Ni-NTA, and 0.5 mol% Texas Red-DHPE. Buffer: 25 mM HEPES, 150 mM NaCl, pH 7.4. Scale bars, 10 μm. Source data are provided as a Source Data file.

(Fig. 6a). The density of FUS LC proteins in the protein-rich domains was estimated to be ~30,000 molecules/μm² based on the previous work[15], which, given that the radius of gyration of FUS LC is approximately 3 nm[46], suggests that FUS LC proteins cover approximately 90% of the membrane surface in the protein-rich region. This coverage suggests that the protein-rich phase in our experiments likely consists of a single layer of closely-spaced, interconnected proteins at the membrane surface. We also observed transmembrane domain coupling when FUS LC proteins were allowed to phase separate on both sides of the membrane at the same time (Fig. 6b, c). Importantly, phase separation of FUS LC relies heavily on pi-pi stacking interactions among tyrosine residues, whereas phase separation of RGG is dominated by electrostatic interactions. Given these differences, the very similar impact of the two proteins on membrane organization suggests that the transmembrane coupling observed here is a general phenomenon that may be applicable to diverse proteins that phase separate at membrane surfaces.

## Discussion

Molecular reconstitution has provided fundamental insights into the mechanisms behind protein phase separation, both in solution[1–4] and at membrane surfaces[5–7,14–18]. However, because existing membrane substrates for in vitro reconstitution provide access to only one side of the membrane, interactions between phase separated regions on opposite sides of a membrane surface have not been investigated previously.

Here we introduce a freestanding planar membrane array as a platform to study membrane-associated protein phase separation on both sides of a membrane surface at the same time. Using this approach, our images reveal coupling between protein-enriched condensates on one side of the membrane and those on the opposite side. In particular, our findings suggest that liquid-like protein assemblies

on membranes, regardless of the identity of proteins, create reorganized lipid regions with reduced lipid mobility, which become coupled to reorganized regions within the opposite leaflet. Notably, transmembrane coupling in our system was highly stable, such that, once they became coupled, membrane-bound protein condensates on opposite sides of the membrane were never observed to separate. The reduction in free energy owing to transmembrane coupling of liquid-ordered lipid domains has been estimated to be approximately 0.016 $k_B$T/nm[2,39]. Given that the coupled domains in our experiments have micrometer dimensions, the energetic barrier to uncoupling would be on the order of ~ $10^4$ $k_B$T, suggesting highly stable coupling. By the same logic, a coupled domain of 20–30 nanometers in diameter should incur a barrier to uncoupling of approximately 10 $k_B$T, significantly above the thermal energy. These arguments suggest that stable coupling could extend to small length scales, relevant to many physiologically important structures. However, the precise scaling between domain size and stability remains to be measured.

From a physical perspective, transmembrane coupling represents a previously unknown mechanism for transferring information across biological membranes, which is independent of membrane-spanning proteins and lipid-lipid immiscibility. In particular, we demonstrate that a protein condensate on one side of the membrane can be detected by a condensate on the other side of the membrane through their mutual influence on the organization of the underlying lipids, a process that does not require a discontinuity in lipid composition, or direct contact between proteins on the two surfaces of the membrane.

From a biological perspective, it is increasingly evident that liquid-like proteins help to organize critical structures and events at biological membranes, from the assembly of cell-cell junctions to the budding of trafficking vesicles[9–11]. Importantly, many such structures require protein-protein interactions on both surfaces of the membrane. In these processes, we speculate that transbilayer coupling of protein condensates could work in concert with other factors including transmembrane proteins, lipid phase separation[47], and membrane curvature to achieve robust transbilayer communication. Determining the potential for synergy between these effects is an interesting subject for future work.

## Methods

### Materials

1,2-dioleoyl-sn-glycero-3-phosphocholine (DOPC), 1,2-dioleoyl-sn-glycero-3-[(N-(5-amino-1-carboxypentyl)iminodiacetic acid)succinyl] nickel salt (DGS-Ni-NTA), 1-palmitoyl-2-{6-[(7-nitro-2-1,3-benzoxadiazol-4-yl)amino]hexanoyl}-sn-glycero-3-phosphocholine (NBD-PC), 1,2-dioleoyl-sn-glycero-3-phospho-L-serine (DOPS), and phosphatidylinositol-4,5-bisphosphate (PI(4,5)P2) were purchased from Avanti Polar Lipids. Texas Red 1,2-dihexadecanoyl-sn-glycero-3-phosphoethanolamine triethylammonium salt (Texas Red-DHPE), BODIPY TR Ceramide, Oregon Green 488 1,2-dihexadecanoyl-sn-glycero-3-phosphoethanolamine (Oregon Green-DHPE), N-(7-nitrobenz-2-oxa-1,3-diazol-4-yl)-1,2-dihexadecanoyl-sn-glycero-3-phosphoethanolamine triethylammonium salt (NBD-DHPE), and 4-(2-hydroxymethyl)-1-piparazineethanesulfonic acid (HEPES) were purchased from Thermo Fisher Scientific. Sodium chloride, sodium tetraborate, hexadecane, squalane, silicone oil AR 20, poly-L-lysine MW 15,000-30,000 (PLL), Atto 488 NHS ester, and Atto 594 NHS ester were purchased from Sigma-Aldrich. Amine-reactive PEG (mPEG–succinimidyl valerate, MW 5000) was purchased from Laysan Bio.

### Plasmids

The plasmid for RGG (pET-RGG) was a gift from Matthew Good, Daniel Hammer, and Benjamin Schuster (Addgene plasmid # 124929; https://www.addgene.org/124929)[21]. The plasmid for FUS LC (RP1B FUS 1-163) was a gift from Nicolas Fawzi (Addgene plasmid # 127192; https://www.addgene.org/127192)[48]. The plasmid for the epsin1

N-terminal homology domain (residues 1-164, pGEX4T2-GST-Thrombin-ENTH) was a gift from Harvey McMahon[36]. The plasmid for the recombinant protein chimera ENTH-RGG was generated by inserting the gene fragment for RGG from the above plasmid (pET-RGG) into the plasmid for ENTH (pGEX4T2-GST-Thrombin-ENTH) by Gibson assembly.

For that, the gene fragment for RGG was PCR amplified using the forward primer, 5′-CCACGGCTTCCTCAGCAGCTATGGAATCCAATCAATCCAATAACGGTGG-3′, and the reverse primer, 5′-CAGACAAGCTGTGACCGTCTTCACTCGAGGCCATCGC-3′.

The plasmid for ENTH (pGEX4T2-GST-Thrombin-ENTH) was PCR amplified using the forward primer, 5′-GTGGCGATGGCCTCGAGTGAAGACGGTCACAGCTTGTCTG-3′, and the reverse primer, 5′-TTGGATTGATTGGATTCCATAGCTGCTGAGGAAGCCG-3′.

The amplified fragments were purified by gel extraction (QIAEX II Gel Extraction Kit, QIAGEN), followed by Gibson assembly using NEBuilder HiFi DNA Assembly Master Mix (New England Biolabs). The Gibson assembly reaction product was then transformed into DH5α bacterial cells, spread on Ampicillin plates, and grown for 16 h. The colonies were screened for successful DNA fragment insertion. The successful creation of ENTH-RGG recombinant protein chimera was confirmed by DNA sequencing.

## Protein expression and purification

Expression and purification of the RGG domain of LAF-1 protein (RGG) was performed as described previously[21]. Briefly, E. Coli BL21(DE3) competent cells were transformed with a plasmid encoding the RGG domain of LAF-1. After transformation, cells were grown in 1 L of 2xYT media for 3-4 h at 37 °C while shaking at 220 rpm until the optical density at 600 nm (OD 600) of the media became reached 0.8, followed by overnight expression induced with 0.5 mM of iso-propyl β-D-1-thiogalactopyranoside (IPTG) at 18 °C while shaking at 220 rpm. Pellets of cells expressing RGG were harvested through centrifugation at 4 °C. Pellets were resuspended in 40 mL buffer containing 20 mM Tris, 500 mM NaCl, 20 mM imidazole, 1% Triton X-100, and one EDTA-free protease inhibitor tablet (Sigma Aldrich) at pH 7.5, and lysed by sonication on ice. To prevent the formation of RGG condensates, all of the following steps were done at room temperature. The cell lysate was clarified by centrifugation at $15,000 \times g$ for 30 min and then incubated with Ni-NTA resin (G Biosciences, USA) for 1 h to achieve binding. Protein-bound Ni-NTA resin was settled in a glass column and washed with a buffer containing 20 mM Tris, 500 mM NaCl, 20 mM imidazole, at pH 7.5. The bound proteins were eluted from the Ni-NTA resin with a buffer containing 20 mM Tris, 500 mM NaCl, 500 mM imidazole, at pH 7.5. Purified proteins were then buffer exchanged into the storage buffer (20 mM Tris, 500 mM NaCl, pH 7.5). Small aliquots of the protein were flash-frozen using liquid nitrogen and stored at −80 °C.

Expression and purification of the low-complexity domain of fused in sarcoma (FUS LC) was carried out according to previous reports[15,45]. In brief, FUS LC was overexpressed in E. Coli BL21(DE3) cells. Cells were grown for 4 h at 37 °C while shaking at 220 rpm until OD 600 reached 0.8, followed by expression induced with 1 mM of IPTG at 37 °C for 3 h. Pellets of cells expressing FUS LC were collected and lysed by sonicating on ice in 40 mL lysis buffer containing 500 mM Tris pH 8.0, 5 mM EDTA, 5% glycerol, 10 mM β-mercaptoethanol, 1 mM phenylmethylsulfonyl fluoride (PMSF), 1% Triton X-100 and one EDTA-free protease inhibitor tablet (Sigma Aldrich). The cell lysates were centrifuged at $125,000 \times g$ for 40 min at 4 °C. Unlike RGG, FUS LC resided in the insoluble fraction after centrifugation. Therefore, the insoluble fraction was kept and resuspended in 8 M urea, 20 mM NaPi pH 7.4, 300 mM NaCl, and 10 mM imidazole. The resuspended sample was centrifuged at $125,000 \times g$ for 40 min at 4 °C. Under denaturing conditions, FUS LC is soluble and resides in the supernatant. This supernatant was mixed with Ni-NTA resin (G Biosciences, USA) for 1 h at 4 °C. The Ni-NTA resin was then settled in a glass column and washed with the above solubilizing buffer. The bound proteins were eluted from the Ni-NTA resin with a buffer containing 8 M urea, 20 mM NaPi pH 7.4, 300 mM NaCl, and 500 mM imidazole. Purified proteins were then buffer exchanged into storage buffer (20 mM CAPS, pH 11) using 3 K Amicon Ultra centrifugal filters (Millipore, USA). Small aliquots of the protein were flash-frozen using liquid nitrogen and stored at −80 °C.

Expression and purification of the recombinant protein chimera ENTH-RGG were carried out as follows. The pGEX4T2-GST-Thrombin-ENTH-RGG plasmid was transformed into E. coli BL21 competent cells (New England Biolabs #C2530). Cells were grown at 30 °C until OD 600 reached 0.8. Protein expression was induced with 0.5 mM IPTG, and cells were shaken at 200 rpm at 30 °C for 6–8 h. The rest of the protocol was carried out at 4 °C. The cells were pelleted from 2 L culture by centrifugation at $4785 \times g$ (Beckman JLA-8.1000) for 20 min. Cells were resuspended in 100 mL lysis buffer (0.5 M Tris-HCl pH 7.2, 10% Glycerol, 5 mM EDTA, 5 mM DTT, 1 mM PMSF) with EDTA free protease inhibitor tablets (1 tablet/50 mL, Roche #05056489001), 1.0% Triton-X100, followed by homogenization with a Dounce homogenizer and sonication (4 × 2000 Joules). The lysate was clarified by ultracentrifugation at $29,535 \times g$ (Beckman JA-25.50) for 30 min. The clarified lysate was then applied to a 10 mL bed volume Glutathione Sepharose 4B (Cytiva #17075601) column, washed with 10 column volumes of lysis buffer with 0.2% Triton X-100 and EDTA-free protease inhibitor tablets (1 tablet/50 mL), followed by washing with 5 column volumes of lysis buffer. The protein was eluted with 15 mM reduced glutathione in lysis buffer, followed by pH adjustment back to the pH before adding reduced glutathione. Protein was concentrated with Amicon Ultra-15 Centrifugal Filters, 10 K MWCO (Thermo Scientific, #UFC901024) down to 4 mL. The GST-tag was cleaved by exchanging the protein into Thrombin digestion buffer (50 mM Tris-HCl, pH 7.2, 10 mM CaCl₂,150 mM NaCl, 5 mM EDTA), using a Zeba Spin Desalting Column (Thermo Scientific #89891), followed by running the column (1 mL bed volume Thrombin resin equilibrated with 10 column volumes of Thrombin digestion buffer), with gentle rocking overnight at 4 °C. The GST-tag and uncut fusion protein were removed by a second Glutathione Sepharose 4B column. Protein was concentrated with Amicon Ultra-15 Centrifugal Filters, 10 K MWCO down to the desired volume. Then, centrifugation at $335,400 \times g$ (Beckman TLA-120.2) was conducted for 10 min at 4 °C to remove the aggregated protein. Small aliquots of the protein were flash-frozen using liquid nitrogen and stored at −80 °C.

## Protein labeling

For visualization, RGG, FUS LC, and ENTH-RGG were labeled with the amine-reactive Atto 488 (or Atto 594) NHS ester. The labeling reaction for RGG and ENTH-RGG took place in its storage buffer (25 mM HEPES, 500 mM NaCl, pH 7.5). Dye was added to the protein in 2-fold stoichiometric excess and allowed to react for 30 min at room temperature. Labeled protein was separated from unconjugated dye using Amicon Ultra 0.5 mL centrifugal filters with MWCO of 3 K (for RGG and FUS LC) or 10 K (for ENTH-RGG). The labeling ratio was measured using UV-Vis spectroscopy. Labeled proteins were dispensed into small aliquots, flash-frozen in liquid nitrogen, and stored at −80 °C. Labeling of FUS LC followed the same process except that the labeling reaction was done in the buffer containing 50 mM HEPES at pH 7.4, and the labeled protein was then exchanged back into the storage buffer (20 mM CAPS, pH 11), divided into small aliquots, flash frozen in liquid nitrogen and stored at −80 °C.

## Freestanding planar lipid membrane formation

First, lipids dissolved in chloroform were mixed in a glass vial and dried under a gentle N₂ stream. The dried lipid film was redissolved in a

mixture of hexadecane and silicone oil (1:1 v/v) to obtain a lipid/oil solution with a total lipid concentration of 3 mM. Each oil was filtered through a 0.2 μm syringe filter (Corning Inc.) before use. The lipid/oil solution was bath-sonicated for 30 min and used for experiments within several hours.

The imaging chamber was assembled by placing a PDMS gasket onto no. 1.5 glass coverslips (VWR). To make a PDMS gasket, 2.1 g of Sylgard™ 184 silicone elastomer base and 0.3 g of curing agent (Dow Corning) were thoroughly mixed and poured into a single well of a 12-well plate, where a cylindrical structure with a diameter of 8 mm was placed at the center to make a circular hole in the PDMS layer. The mixture was then incubated at 45 °C for at least 3 h for curing. Coverslips were cleaned with 2% v/v Hellmanex III (Hellma Analytics) solution, rinsed thoroughly with deionized water, and dried under a gentle $N_2$ stream before use. The coverslip was then passivated with a layer of PLL-PEG, which was synthesized as described previously[49]. In each imaging chamber, 15 μL of PLL-PEG solution was added, followed by repetitive rinsing with the experimental buffer using a pipette after 20 min of incubation. A total of 150 μL of the aqueous buffer was then added into the PLL-PEG passivated imaging chamber. After that, 2 μL of lipid/oil solution was gently dropped and spread on the air-aqueous buffer interface. After several minutes, a hexagonal TEM grid made of gold (G150HEX Au, Gilder Grids), which was hydrophobically coated by incubating in 1-dodecanethiol solution (5 mM dissolved in ethanol) overnight before use, was gently placed on the air-oil interface using tweezers. The grid was placed there for several minutes to create a thin oil film within the grid holes. Then, the grid was submerged into the aqueous buffer using a syringe needle to place it on the PLL-PEG-coated glass surface. The thickness of the oil film decreased as the oil drained out, and spontaneous adhesion of two lipid monolayers occurred, resulting in a suspended planar lipid bilayer. Proteins were added to the aqueous buffer above the lipid bilayer (Supplementary Fig. 1).

In experiments requiring a spacer, two small strips of Scotch® Magic™ Tape (3 M), each a few millimeters wide, were adhered to the coverslip surface. The space between these strips was a few millimeters, slightly less than the diameter of the grid. The same procedure was followed to coat the grid with the lipid oil solution to form the membranes. When the grid was submerged in the aqueous buffer, care was taken to settle it on top of the strips of tape so that it bridged the gap between them, such that the protein solution could flow beneath the grid.

## Microscopy

A spinning disk confocal microscope (SpinSR10, Olympus) equipped with a Hamamatsu Orca Flash 4.0V3 sCMOS Digital Camera was used to visualize samples. 20X objective (1-U2M345, Olympus), 1.40 NA/40X oil immersion objective (1-UXB220, Olympus), and 1.50 NA/100X oil immersion objective (1-UXB170, Olympus) were used for visualization. Laser wavelengths of 488 and 561 nm were used for excitation. FRAP was performed using the Olympus FRAP unit 405 nm laser and 1.50 NA/100X oil immersion objective (1-UXB170, Olympus).

## Image analysis

ImageJ was used for image analysis. Contrast and brightness were kept the same for time-lapse image sets for all images. For all cases, fluorescence intensity values were measured in unprocessed images.

For FRAP analysis, the FRAP Profiler plugin for ImageJ was used. Fluorescence recovery of regions with 2–5 μm diameter was measured over time, and intensity values were normalized to the maximum pre-bleach and minimum post-bleach values.

For the fusion dynamics of two coalescing membrane-protein composite regions observed in the lipid channel, aspect ratio values were measured using the "Analyze Particles" function.

To measure partition coefficients ($K_P$) of Texas Red DHPE under different NaCl concentrations, fluorescence intensity values (background subtracted) of two adjacent brighter and dimmer regions were used to avoid the effect of local variations in intensity.

## Statistics and reproducibility

For all experiments producing micrographs, each experiment was repeated independently at least three times, with similar results. Statistical analysis was carried out using a Student's unpaired $t$ test, with a two-tailed distribution.

## Reporting summary

Further information on research design is available in the Nature Portfolio Reporting Summary linked to this article.

## Data availability

The data supporting the findings of this study are available from the corresponding author upon request. The source data for Figs. 1f, 2c, 3c, g, 4b, c, 5d, f–h, 6c, and Supplementary Fig. 7 are provided as a Source Data file. Source data are provided with this paper.

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

## Acknowledgements

This research was supported by the NIH through grant R35GM139531 (J.C.S), the NSF DMS through grant 1934411 (J.C.S), and the Welch Foundation through grant F-2047 (J.C.S).

## Author contributions

Y.L., S.P. and J.C.S. designed experiments. Y.L., S.P., F.Y., C.C.H., L.W. and E.M.L. performed experiments and data analysis. Y.L. and J.C.S wrote and edited the manuscript. S.P., F.Y., C.C.H., L.W., E.M.L. and S.Q.C. participated in writing and reviewing the manuscript.

## Competing interests

The authors declare no competing interests.
