## [Peer review file · Nature Communications]

REVIEWER COMMENTS

Reviewer #1 (Remarks to the Author):

Authors of this manuscript reports a phenomenon that protein condensates formed on both side of lipid membranes are coupled via modifying the lipid ordering. Previous studies have shown that adsorption or binding of proteins to one side of lipid membranes can modify the behaviors, including ordering, of lipid phases. It is also known that microscopic lipid phase separation in one leaflet can induce phase separation in the opposing leaflet via lipid interleaflet coupling. Therefore, it has been expected that the protein domains formed on both sides of the leaflets should be coupled. In this manuscript, the authors reported an elegant experimental setup to demonstrate this phenomenon. They formed free standing lipid membranes on TEM grids and used spacers to allow protein binding (via his-tags) to both leaflets of the bilayer.

The authors have provided a well-designed and carefully executed experimental setup to demonstrate interleaflet coupling of protein condensate domains formed on both sides of lipid membranes. The data analysis was thorough and rigorous, making this a solid paper.

1. It would be helpful if the authors could provide information on the density of proteins in the protein-rich and protein-poor domains, as well as the respective surface coverages of proteins in both domains, considering the large size of the proteins.
2. While the authors have presented observations that suggest protein condensates increase lipid ordering, the depletion of lipid probes in the protein-rich domains does not necessarily mean an increase in lipid ordering. Several different factors could have resulted in a similar partition of lipid probes in the lipid membrane. For example, the lipids underneath the protein condensates could be "pinned" locally by weak dipolar electrostatic interactions with the proteins (see Zhang et al. PNAS 102, 9118, 2005). It is also possible that the crowing effect from the proteins modifies the membrane hydration locally (see Mangiarotti et al. Biochim Biophys Acta Biomembr, 1863(12):183728, 2021), or physically hinders the diffusion of lipids, or mechanically compresses the lipid bilayer locally (Yim et al. Phys Rev Lett. 2006; 96(19): 198101). The authors could strengthen their conclusions by providing additional data to support their claims.
3. The manuscript is well written generally, but there are a few typos that should be corrected. For example, on SI page 3, the sentence "To prepare First, lipids dissolved in chloroform were..." is incomplete. Some sentences in the manuscript could also be revised for easier understanding. For example, the sentence "We also observed that like phases, either protein-rich or protein-poor, fused and re-rounded upon contact, such that the phase separation coarsened over time" could be rephrased for clarity.

Reviewer #2 (Remarks to the Author):

This submission experimentally explores an interesting membrane model, in which membranes span apertures within an electron microscopy grid, leaving large (100 micron) regions with unsupported membranes. These closely resemble black lipid membranes most frequently used in electrophysiology, and contain some oil. These authors conjugate his-tagged LAF-1 to these membranes via Ni lipids, and observe 2D protein-rich condensates, similar to other studies in this area. This study extends upon past work in a few areas. First, they are able to separately conjugate each side of the membrane, enabling and they observe that 2D surface phases tend to align across-leaflets. They use the language of trans bilayer coupling to describe these measurements, but I think it might be more (or at least equally) appropriate to describe these as 3 coexisting surface phases. They also find that their lipid probes are excluded from 2D phases, and that the viscosity (as represented by the diffusion coefficient of lipid probes) differs between the 2 or 3 phases observed.

I find the experimental results presented to be interesting, but they for the most part report on results that are expected given the experimental conditions probed and what is already known about these systems. The asymmetric addition of protein and the observation that phases are in register is a novel result, but I do not think the authors go far enough to be able to claim that their proposed mechanism is relevant. In addition to addressing the comments below, I recommend that a revision, if requested, dramatically tone down the language regarding experimental results supporting a particular model. For example, the conclusion stated in the title is not supported by the data presented.

Specific comments:

1. The authors should indicate Ni lipid concentration and the presence of oil in the results text and not just in methods/figure captions. These are both distinguishing features from other related studies. 15% or 25% Ni lipid is a very large mole%.
2. The phase diagram for this phase transition (presented somewhat in Fig 1) will depend on ionic strength and the mole% of Ni lipid in the membrane and also likely on the concentration of protein in solution. It is overly simplistic to imply that observing different surface fractions of phases is enough to distinguish the various contributions of each of these pieces. Why not map the actual phase diagram instead? Or at least present these results in this context without collecting all of the data required for a more detailed description. As is, very little is actually concluded from this section.
3. The images shown in 2a and 2b are nowhere near a critical composition (the transition is clearly first order). This should be clarified, since the text seems to suggest that these are critical.
4. Regarding the lipid exclusion – At first glance this result is in disagreement with those of other groups (and of this group in past studies (e.g. 10.1073/pnas.2017435118)) using other membrane preparations and membrane coupled proteins -- and therefore requires more explanation. It seems that this could be due to the very large mole fraction of Ni lipids used (15 mol% in Fig 4)? In the pictures shown, the protein rich phase makes up $\sim 1/4$ of the membrane area, then presumably Ni lipids probably make up

~50% of the lipids in the protein rich region? This is not a minor fraction. I did not see any numbers presented regarding partitioning coefficients so can not tell if this simple explanation is sufficient. I think the authors should be explicit about the likely Ni lipid concentration within protein rich phases in the discussion of these results. It also should be made clear that this result is not particularly general as suggested in the text.

5. Regarding the observations that different viscosities are observed in the different phases, I believe that this is an expected result given the large fraction of Ni lipids that are interacting strongly with protein. Even outside of this, the system is clearly separated into multiple surface phases, and it is expected that these phases would have different physical properties such as viscosity. Even in the absence of a large Ni lipid concentration, I would expect frictional coupling with protein would impact lipid mobility. It also follows that there would be different viscosities in the systems with three phases (protein-depleted phase, protein-enriched phase on one side, protein-enriched phases on both sides). While it is nice to see this expectation validated experimentally, it does not provide evidence for a specific coupling mechanism, as stated in the text.

6. I strongly disagree with the conclusion that coupling need be accomplished through “entropic” ordering interactions with the chains. There are many possible explanations that would involve energetic contributions to the free energy treating the membrane as an elastic sheet (as is typically done) – e.g. bending rigidity, spontaneous curvature coupling, etc. There is a ton of theory literature in this area. Of course entropy is also involved in the free energy but there are many contributions to this too. I would find the results of this study to be much more unexpected if no apparent coupling were observed, since there are so many ways in which coupling could be expected to arise. This is not to say that the experimental results are not interesting, but framing them as “reducing entropy” is not particular informative and also misleading – certainly this has not been shown here decisively.

Relevant theory has been done on a related system (minus the asymmetry present here): e.g. [10.1073/pnas.2103401118](https://doi.org/10.1073/pnas.2103401118), [10.1088/1367-2630/ac320b](https://doi.org/10.1088/1367-2630/ac320b). It seems like these (or similar) should be cited along with the past experimental studies mentioned, but may also be helpful in developing a more rigorous theoretical explanation.

Reviewer #3 (Remarks to the Author):

Reviewer Summary

In Lateral compression of lipids drives transbilayer coupling of liquid-like protein condensates, Lee et al. develop methodology to investigate coupling membrane-proximal condensates located on either side of a synthetic membrane. They stretch planar membranes over an EM grid and flow an aqueous solution with phase-separating proteins (RGG, FUS) over one or both sides of the bilayer. Proteins may bind to

lipids tagged Ni-DGS lipids. They demonstrate that condensates form on these membranes, – and do so at concentrations and interactions strengths far weaker than those required to phase-separate in bulk; the authors chart a phase – diagram of this membrane-condensate system.

Condensates that are allowed to interact with lipids on both sides of the membrane are found to be “coupled”: fusion-like events co-localize these transbilayer condensates. The authors propose an “entropic compression” mechanism underlying this phenomena, whereby condensates increase the local ordering of membrane lipids, and that the entropic penalty for doing such can be abrogated by colocalized condensates in one-domain, as opposed to two. Results are demonstrated in simple phase-separating protein systems LAF-RGG and FUS.

This manuscript presents a first step at understanding how condensates interact with both sides of a lipid membrane. The methodology and results pertaining to bilayer coupling are new and interesting. I am, however, concerned with the interpretation and presentation of results pertaining to the proposed mechanism of transbilayer coupling.

Major Comments

1) The authors suggest an “entropic” coupling mechanism, whereby a protein condensate increases the order of the underlying lipids. This is an interesting possibility, but the claim is stated too confidently, and not adequately tested. I suggest that the authors further verify some of their results, and temper the confidence with which they claim this mechanism in the results and discussion.

i) This putative mechanism relies on the lipids underlying the condensate becoming more ordered, yet the authors only test that the condensates exclude disordered dye. Does an ordered-phase dye partition into the lipid phases underlying these condensates? If possible, the authors should perform experiments in the coupled system with ordered phase dye to see if this partitions into the condensate-rich phase as expected.

ii) condensates could interact with the dye. One could expect that condensates could repel the dye molecule, depleting it from the phase without needing any ordering mechanism. Slower kinetics would then be related to an enrichment of protein-interacting DGS-Ni-NTA lipids in this phase. Do condensates that form in bulk solution interact with the dye at all?

iii) Dyes could have differential partitioning in DOPC vs DGS. If this is the case, depletion of dyes (and perhaps slower diffusion) within the droplets could be explained by the enrichment of DGS-Ni-NTA lipids in the domain. The authors should check that this is not the case

iv) This mechanism implies that a membrane composed of a lipid that is more ordered would present less of an effect, as there is less entropy to be “lost”. Can the authors perform this experiment in a membrane composed of ordered-phase lipids, and see if condensates still couple across the bilayer?

2) The coupled-bilayer experiments suggest coexistence of three distinct quasi-2D liquid phases, – protein rich on both sides, protein rich on one side, and protein poor. This is reminiscent of three-phase coexistence in asymmetric, multicomponent lipid membranes (vide Collins & Keller 2008) and could, in spirit, be the same phenomena.

i) Can the authors comment on whether this is a correct interpretation? If so, why avoid calling this phenomenon three-phase coexistence? If not, why?

ii) asymmetric membranes grant a bilayer some coupling across leaflets, as can curvature and membrane spanning proteins. Can the authors comment on the magnitude of these effects compared to their proposed mechanism? In the Discussion the authors calculate the order of these effects, – would this be a small correction to the other coupling mechanisms, or a dominant effect? Real cells often have different condensates on different sides of the membrane presenting differences in possibly the scale and the sign of these effects. The authors should temper their claims about transbilayer communication via this mechanism alone.

iii) Is the membrane symmetric in DGS-Ni-NTA concentration? If so, the domains should (after a very long time) all be coupled across the membrane. Should I interpret the effects as a kinetic effect but indicative of equilibrium phase behavior?

4) In Figure 2 the authors claim that bilayer coupling increases the critical temperature of membrane-bound condensates. I find the presentation of this claim troubling, as it conflates critical temperatures with transition temperature. The mixtures do not show critical-like fluctuations, and are likely not near a critical point.

i) The authors should clarify the language involved in this statement (e.g change “critical temperature” to “transition temperature”) to be less suggestive of being near a critical point.

ii) Another concern is that this conflates surface behavior with bulk behavior, when the transition on the membrane could be a qualitatively different transition (vide Rouches et al PNAS 2021, Zhao et al NJP 2021). The authors should draw consider whether they are changing the bulk phase boundary, the surface phase boundary, or both.

5) Related: The phase-diagram the authors draw in Figure 2 is not in terms of a conserved quantity. The only conserved quantity here is the DGS-Ni-NTA concentration. Is there a way to measure the DGS concentration in each phase? If so the authors should draw a phase-diagram in this space, which would be more physically interpretable.

i) related: Figures 1G,H are nearly a phase diagram in the correct coordinates, traversing across a tie-line but chop the space up into various cases corresponding to surface-dilute (case 1), surface-dense (case 4), surface-coexistence (case 2 & 3). Why divide the space up into these categories and not measure the area-fraction of the two phases?

6) The authors cite that Laf-RGG phase separates in 3D at near 10 micromolar concentration. The cited paper (Elbaum-Garfinkle 2015) shows that, depending on salt concentration, this can be reduced to ~ one micromolar. Can the authors present their “phase diagram” of a unary LAF-RGG solution, perhaps in the supplement?

Minor comments

1) The authors say that they “increase the strength of protein membrane interactions” by increasing the amount of DGS-Ni-NTA lipid in the membrane. This seems more like they increase the number of these interactions, while holding their strength fixed. Can the authors clarify this language?

2) The authors write “... the cooperative relationship between protein binding and phase separation...”: It is very unclear what they mean here.

3) The authors use the term “simultaneously” to mean “co-localized”, or something else. Do the domains really form at the same time?

4) An implicit claim behind the authors interpretation is that, although the condensates “communicate” with one another, there is no additional stabilization of condensates across the bilayer. Is it possible that this coupling can stabilize condensates in regimes where they don’t form on membranes with only one accessible bilayer? I would guess this would be a small regime, and maybe not accessible.

Typographical

On the second paragraph of results and discussion, the authors write “Further RGG is high soluble” should read “highly soluble”

Response to Reviews for the submission NCOMMS-23-09926-T

Title

Transmembrane coupling of liquid-like protein condensates (*Title altered in response to reviewer suggestions, detailed below*)

Authors

Yohan Lee, Sujin Park, Feng Yuan, Carl C. Hayden, Liping Wang, Eileen M. Lafer, Siyoung Q. Choi and Jeanne C. Stachowiak

We thank the reviewers for their thorough review of our work and for their helpful comments and suggestions. Below we provide responses to each of the reviewers' comments and describe the actions we have taken to address them. The changes we have made to the text are highlighted in the revised manuscript. We hope the reviewers will agree that we have significantly strengthened the manuscript by responding to their comments.

Reviewer Comments:

Reviewer #1:

Authors of this manuscript reports a phenomenon that protein condensates formed on both sides of lipid membranes are coupled via modifying the lipid ordering. Previous studies have shown that adsorption or binding of proteins to one side of lipid membranes can modify the behaviors, including ordering, of lipid phases. It is also known that microscopic lipid phase separation in one leaflet can induce phase separation in the opposing leaflet via lipid interleaflet coupling. Therefore, it has been expected that the protein domains formed on both sides of the leaflets should be coupled. In this manuscript, the authors reported an elegant experimental setup to demonstrate this phenomenon. They formed freestanding lipid membranes on TEM grids and used spacers to allow protein binding (via his-tags) to both leaflets of the bilayer.

The authors have provided a well-designed and carefully executed experimental setup to demonstrate interleaflet coupling of protein condensate domains formed on both sides of lipid membranes. The data analysis was thorough and rigorous, making this a solid paper.

Author response: We appreciate the reviewer's careful reading and understanding of our manuscript.

Comments:

1. It would be helpful if the authors could provide information on the density of proteins in the protein-rich and protein-poor domains, as well as the respective surface coverages of proteins in both domains, considering the large size of the proteins.

Action taken: Based on our previous work¹, the number of FUS LC proteins in protein-rich domains per membrane area was estimated to be $\sim 30,000$ molecules/ μm^2 through single

molecule calibration. This value was obtained from giant unilamellar vesicles (GUVs) (composition: 83 mol% POPC, 15 mol% DGS-Ni-NTA, 2 mol% DP-EG10-biotin) incubated with 1 μM of his-FUS LC (10% Atto-488 labeled) in 25 mM HEPES, 150 mM NaCl, pH 7.4 buffer. Meanwhile, the radius of gyration (R_g) of FUS LC has been estimated as 3 nm². Based on this value, the projected area of FUS LC on a two-dimensional surface should be approximately 30 nm² (πR_g^2). Using this estimate, 30,000 FUS LC molecules/ μm^2 corresponds to a coverage of the membrane surface of approximately 90% in the protein-rich phase. This density suggests that the protein-rich phase on membrane surfaces likely consists of a single layer of closely-spaced, interconnected proteins.

We have added the following sentence in the revised manuscript to provide information on the protein density to the readers:

“The density of FUS LC proteins in the protein-rich domains was estimated to be $\sim 30,000$ molecules/ μm^2 based on the previous work¹, which, given that the radius of gyration of FUS LC is approximately 3 nm², suggests that FUS LC proteins cover approximately 90% of the membrane surface in the protein-rich region. This coverage suggests that the protein-rich phase in our experiments likely consists of a single layer of closely-spaced, interconnected proteins at the membrane surface.”

We also quantified the area fractions of protein-rich domains for his-RGG at each NaCl concentration (100 mM, 150 mM, and 200 mM) and DGS-Ni-NTA concentration (5 mol%, 15 mol%, and 20 mol%). These data are shown in Supplementary Figure 2, which is copied below and has been added to the Supplementary Information of our revised submission. These data clearly show that the area fraction, while heterogeneous, increases, as expected, with increasing DGS-Ni-NTA concentration and decreasing NaCl concentration.

Supplementary Figure 2. Area fraction of protein-rich regions as a function of NaCl concentration and DGS-Ni-NTA concentration. Total 20 lipid membranes were analyzed from 2-3 independent experiments for each condition. Bars represent the average area fraction for

each case. Membrane composition: 75-95 mol% DOPC, 5-25 mol% DGS-Ni-NTA. 1 μ M of his-RGG labeled with Atto 488 were used.

The following statement was added to the main text to introduce this figure:

“Similarly, the fraction of the membrane area covered by the protein-rich phase increased with decreasing NaCl concentration and increasing Ni-NTA concentration (Supplementary Fig. 2).”

2. While the authors have presented observations that suggest protein condensates increase lipid ordering, the depletion of lipid probes in the protein-rich domains does not necessarily mean an increase in lipid ordering. Several different factors could have resulted in a similar partition of lipid probes in the lipid membrane. For example, the lipids underneath the protein condensates could be “pinned” locally by weak dipolar electrostatic interactions with the proteins (see Zhang et al. PNAS 102, 9118, 2005). It is also possible that the crowding effect from the proteins modifies the membrane hydration locally (see Mangiarotti et al. Biochim Biophys Acta Biomembr, 1863(12):183728, 2021), or physically hinders the diffusion of lipids, or mechanically compresses the lipid bilayer locally (Yim et al. Phys Rev Lett. 2006; 96(19): 198101). The authors could strengthen their conclusions by providing additional data to support their claims.

Action taken: We have revised our discussion in the main text to provide several possible factors for the lipid probe depletion in the protein-rich domains that we observed, citing the papers mentioned by the reviewer (Yim et al. Phys Rev Lett. 2006; 96(19): 198101 / Mangiarotti et al. Biochim Biophys Acta Biomembr, 1863(12):183728, 2021). We cited “Zhang et al. PNAS 102, 9118, 2005”, one of the papers the reviewer mentioned, in the main text where we were explaining the different lipid mobility observed by FRAP, because we found this literature more related to this phenomenon (Please refer to comment no. 5 from the Reviewer #2). The revised contents in the main text are as follows (newly added text in red):

“Why are diverse probe lipids excluded from the protein-enriched phase? Notably, the conjugation of a fluorophore to a lipid substantially increases its molecular weight, creating a bulky amphiphile that often disrupts the packing of other membrane lipids. As a result, most probe lipids are known to be excluded from liquid-ordered and solid-like membrane phases^{3,4}. This reasoning suggests that **local ordering of the membrane lipids due to assembly of protein condensates on membrane surfaces could explain probe lipid exclusion, similar to the previously reported rearrangement of lipid packing owing to the adsorption of proteins with multiple binding sites**⁵. Notably, the term ordered lipids does not refer to phase-separated lipid regions composed of saturated lipids and sterols, as reported elsewhere in the literature^{3,4}. Instead, we use the term ‘lipid ordering’ to indicate that the conformational freedom of the lipids is locally reduced due to the presence of protein condensates. **Interestingly, we observed that several probe lipids known to partition into liquid-ordered phases (Oregon Green-DHPE and NBD-DHPE), along with those normally associated with liquid-disordered phases (Texas Red-DHPE, BODIPY TR-**

Ceramide, and NBD-PC), were excluded from the protein-rich regions. This result could suggest that protein condensates create an ultra-dense lipid region with a high degree of lipid packing that excludes a larger variety of probe lipids than has been previously observed. However, probe exclusion could also arise from other effects including protein-lipid interactions, or a modified degree of lipid hydration owing to the presence of proteins⁶. Meanwhile, probe lipid exclusion is unlikely to arise from the enrichment of DGS-Ni-NTA lipids in protein-rich regions, as probe lipid exclusion remained constant across a range of DGS-Ni-NTA concentrations (5, 10, 15 mol%) (Supplementary Figs. 6-7). Note that in our past work with GUVs, the head-labeled probe lipid, Texas Red-DHPE was enriched, rather than depleted in the protein-rich regions of the membrane¹. While the reasons for this difference remain unclear, it is possible that the higher membrane tension in suspended membranes, relative to GUVs⁷, permits greater shifts in lipid packing upon protein condensation.”

3. The manuscript is well written generally, but there are a few typos that should be corrected. For example, on SI page 3, the sentence "To prepare First, lipids dissolved in chloroform were..." is incomplete. Some sentences in the manuscript could also be revised for easier understanding. For example, the sentence "We also observed that like phases, either protein-rich or protein-poor, fused and re-rounded upon contact, such that the phase separation coarsened over time" could be rephrased for clarity.

Action taken: We appreciate the referee pointing out these issues. We deleted the phrase “To prepare” to fix the sentence on SI page 3, so that the revised sentence became as follows:

“First, lipids dissolved in chloroform were...”

Also, we revised the sentence "We also observed that like phases, either protein-rich or protein-poor, fused and re-rounded upon contact, such that the phase separation coarsened over time" as follows for clarity:

“We also observed fusion and re-rounding upon contact between different protein-rich domains (Fig. 1b,c) and different protein-poor domains (Fig. 1d), leading to the domain coarsening over time (Supplementary Movie. 1,2).”

Reviewer #2:

This submission experimentally explores an interesting membrane model, in which membranes span apertures within an electron microscopy grid, leaving large (100 micron) regions with unsupported membranes. These closely resemble black lipid membranes most frequently used in electrophysiology, and contain some oil. These authors conjugate his-tagged LAF-1 to these membranes via Ni lipids, and observe 2D protein-rich condensates, similar to other studies in this area. This study extends upon past work in a few areas. First, they are able to separately

conjugate each side of the membrane, enabling and they observe that 2D surface phases tend to align across-leaflets. They use the language of transbilayer coupling to describe these measurements, but I think it might be more (or at least equally) appropriate to describe these as 3 coexisting surface phases. They also find that their lipid probes are excluded from 2D phases, and that the viscosity (as represented by the diffusion coefficient of lipid probes) differs between the 2 or 3 phases observed.

I find the experimental results presented to be interesting, but they for the most part report on results that are expected given the experimental conditions probed and what is already known about these systems. The asymmetric addition of protein and the observation that phases are in register is a novel result, but I do not think the authors go far enough to be able to claim that their proposed mechanism is relevant. In addition to addressing the comments below, I recommend that a revision, if requested, dramatically tone down the language regarding experimental results supporting a particular model. For example, the conclusion stated in the title is not supported by the data presented.

Action taken: We appreciate the reviewer's careful reading and understanding of our manuscript.

In our revised manuscript we note that three different levels of intensity, resulting from transbilayer coupling in our system, can also be described as the coexistence of three phases. In particular, we added the following sentence in the revised manuscript where we were explaining our observations of three levels of intensity:

“Note that this observation can also be described as the coexistence of three different phases⁸.”

We are pleased that the reviewer recognizes the major novel contribution of our work, which is that protein-rich regions on one side of the membrane were observed to come into register, or couple, with protein-rich regions on the opposite side of the membrane.

We agree that there is much more to be learned about the physical mechanisms responsible for this coupling, which likely extend beyond lipid compression. Therefore, we have now shortened the title to, “**Transmembrane coupling of liquid-like protein condensates**”, which does not imply precise knowledge of the underlying mechanisms.

Additionally, we have edited our statements in the discussion to include a range of possible mechanisms, suggesting that future work will be required by the community to fully understand why coupling occurs. The modified text is quoted below:

“It is well known that ordered lipid domains in opposing membrane leaflets frequently become coupled, or registered, by various mechanisms including line tension, membrane fluctuation, and spontaneous curvature⁸⁻¹². Transmembrane coupling of protein condensates could in principle arise from lipid domain coupling. Such coupling could be triggered by the local ordering of lipids, which could be accomplished by altering lipid packing, lipid composition, or membrane curvature, each via protein-lipid interactions (Fig. 5a). Given the diversity of factors that could contribute to our findings, future work by the

community, including theoretical studies^{13,14}, will be needed to isolate the specific mechanisms responsible for transmembrane coupling of protein condensates.”

Comments:

1. The authors should indicate Ni lipid concentration and the presence of oil in the results text and not just in methods/figure captions. These are both distinguishing features from other related studies. 15% or 25% Ni lipid is a very large mole%.

Action taken: We have stated that a small amount of oil might be present in our lipid membrane, and that we also observed a similar lipid probe exclusion using solvent-free membranes, as follows in the manuscript:

“To check if a small amount of residual oil (hexadecane) in our membrane had an effect on lipid probe exclusion, we replaced hexadecane (C₁₆H₃₄) with squalane (C₃₀H₆₂), which has a longer hydrocarbon chain that has been shown to greatly reduce the amount of trapped oil in the bilayer, leading to membranes with a very low solvent content^{15–17}. We observed a very similar depletion of the probe lipid from the protein-enriched regions using squalane (Supplementary Fig. 4), suggesting that depletion of the probe lipid cannot be explained by inclusion of oil in the bilayer.”

We have added the DGS-Ni-NTA concentration used in our experiments in the revised manuscript as follows (newly added text in red):

“We added RGG with an N-terminal histidine-tag (his-RGG), labeled with Atto 488, to freestanding planar membranes containing DGS-Ni-NTA lipids (5-25 mol%).”

We agree that the fraction of Ni-NTA used in our studies is high. In response to a similar concern from another reviewer, we examined the impact of using lower fractions of Ni-NTA. We observed that the percentage of DGS-Ni-NTA can be lowered to 5 mol% without substantially changing the observed phenomena, Supplementary Figure 6 (shown below). Also, we found that lipid probe partitioning is not significantly affected by the DGS-Ni-NTA content of the membrane, Supplementary Figure 7 (shown below).

Supplementary Figure 6. Lipid probe exclusion from protein-rich regions with low DGS-Ni-NTA concentration in the membrane. Membrane composition: 95 mol% DOPC, 5 mol% DGS-

Ni-NTA with 0.5 mol% Texas Red DHPE. Buffer: 25 mM HEPES, 100 mM NaCl, pH 7.4. 1 μ M of his-RGG labeled with Atto 488 was used. Scale bars, 10 μ m.

Supplementary Figure 7. Partition coefficients (K_p) of lipid probe as a function of different DGS-Ni-NTA concentrations in the membrane. Red bars indicate the average K_p values for each case. Membrane composition: 75-95 mol% DOPC, 5-25 mol% DGS-Ni-NTA with 0.5 mol% Texas Red DHPE. 1 μ M of his-RGG was used. Brackets show statistically significant comparisons using an unpaired, two-tailed Student's t test. (n.s. indicates a difference that was not statistically significant.)

The following statement was added to the main text to introduce Supplementary Figure 6 and 7:

“Meanwhile, probe lipid exclusion is unlikely to arise from the enrichment of DGS-Ni-NTA lipids in protein-rich regions, as probe lipid exclusion remained constant across a range of DGS-Ni-NTA concentrations (5, 10, 15 mol%) (Supplementary Figs. 6-7).”

2. The phase diagram for this phase transition (presented somewhat in Fig 1) will depend on ionic strength and the mole% of Ni lipid in the membrane and also likely on the concentration of protein in solution. It is overly simplistic to imply that observing different surface fractions of phases is enough to distinguish the various contributions of each of these pieces. Why not map the actual phase diagram instead? Or at least present these results in this context without collecting all of the data required for a more detailed description. As is, very little is actually concluded from this section.

Action taken: We acknowledge the referee's concern that the diagram in Figure 1h is not a traditional phase diagram. The main message we want to present to the reader with this diagram is that the fraction of the membrane area covered by the protein-rich phase increases with increasing Ni-NTA concentration and decreasing NaCl concentration. This is the behavior

expected for phase separation of the RGG domain in solution¹⁸, and we are simply confirming that this behavior is maintained when RGG binds to the membrane surface. To acknowledge the referee's concern, we have been careful not to refer to Figure 1h as a phase diagram.

Further to make these data more informative for the reader, we used them to calculate the fraction of the membrane surface covered by the protein-rich phase at each combination of DGS-Ni-NTA and NaCl, Supplementary Figure 2 (shown below). Here we see that the area fraction of the protein-rich phase, while somewhat heterogeneous, generally increases with increasing DGS-Ni-NTA concentration and NaCl concentration.

Supplementary Figure 2. Area fraction of protein-rich regions as a function of NaCl concentration and DGS-Ni-NTA concentration. Total 20 lipid membranes were analyzed from 2-3 independent experiments for each condition. Bars represent the average area fraction for each case. Membrane composition: 75-95 mol% DOPC, 5-25 mol% DGS-Ni-NTA. 1 μ M of his-RGG labeled with Atto 488 were used.

The following statement was added to the main text to introduce this figure:

“Similarly, the fraction of the membrane area covered by the protein-rich phase increased with decreasing NaCl concentration and increasing Ni-NTA concentration (Supplementary Fig. 2).”

3. The images shown in 2a and 2b are nowhere near a critical composition (the transition is clearly first order). This should be clarified, since the text seems to suggest that these are critical.

Action taken: We have changed the wording from “critical temperature” to “transition temperature” as the reviewer pointed out.

4. Regarding the lipid exclusion – At first glance this result is in disagreement with those of other

groups (and of this group in past studies (e.g. 10.1073/pnas.2017435118)) using other membrane preparations and membrane coupled proteins -- and therefore requires more explanation. It seems that this could be due to the very large mole fraction of Ni lipids used (15 mol% in Fig 4)? In the pictures shown, the protein rich phase makes up ~1/4 of the membrane area, then presumably Ni lipids probably make up ~50% of the lipids in the protein rich region? This is not a minor fraction. I did not see any numbers presented regarding partitioning coefficients so cannot tell if this simple explanation is sufficient. I think the authors should be explicit about the likely Ni lipid concentration within protein rich phases in the discussion of these results. It also should be made clear that this result is not particularly general as suggested in the text.

Action taken: As the reviewer pointed out, the lipid probe exclusion result in the present work differs somewhat from our recently published work¹. In our past work, Texas Red-DHPE, a head-labeled lipid probe, was slightly enriched in protein-rich regions in giant unilamellar vesicles (GUVs), whereas BODIPY TR-Ceramide, a tail-labeled lipid probe, was depleted in protein-rich regions. Therefore, our present work and the past work show opposite partitioning results for the Texas Red-DHPE lipid probe, while there is agreement between the past and present work concerning partitioning of BODIPY TR-Ceramide. We do not fully understand the reasons for the different partitioning of Texas Red-DHPE in GUVs (past work) versus black lipid membranes (BLMs) (present work). BLMs, in general, have higher membrane tension than GUVs, suggesting that the area per lipid for BLMs is larger than that for GUVs⁷. Given this difference in area per lipid, when protein condensates form on the membrane, the resulting differences in lipid packing between the protein-rich regions and the protein-poor regions might be more significant in the case of BLMs, owing to the initially larger area per lipid in BLMs compared to GUVs. We have added the following sentences in the revised manuscript to note the discrepancy with our past results and to suggest possible explanations:

“Note that in our past work with GUVs, the head-labeled probe lipid, Texas Red-DHPE was enriched, rather than depleted in the protein-rich regions of the membrane¹. While the reasons for this difference remain unclear, it is possible that the higher membrane tension in suspended membranes, relative to GUVs⁷, permits greater shifts in lipid packing upon protein condensation.”

Also, we examined the impact of DGS-Ni-NTA concentration on the extent of lipid probe exclusion by calculating the partition coefficient (K_p) of the lipid probe as a function of DGS-Ni-NTA. This analysis revealed that the partition coefficient of Texas Red-DHPE was not a function of DGS-Ni-NTA concentration, Supplementary Figure 7 (below). These findings suggest that lipid probe exclusion cannot be explained by enrichment of DGS-Ni-NTA in protein-rich regions of the membrane.

Supplementary Figure 7. Partition coefficients (K_p) of lipid probe as a function of different DGS-Ni-NTA concentrations in the membrane. Red bars indicate the average K_p values for each case. Membrane composition: 75-95 mol% DOPC, 5-25 mol% DGS-Ni-NTA with 0.5 mol% Texas Red-DHPE. 1 μ M of his-RGG was used. Brackets show statistically significant comparisons using an unpaired, two-tailed Student's t test. (n.s. indicates a difference that was not statistically significant.)

The following statement was added to the main text to introduce Supplementary Figure 7:

“Meanwhile, probe lipid exclusion is unlikely to arise from the enrichment of DGS-Ni-NTA lipids in protein-rich regions, as probe lipid exclusion remained constant across a range of DGS-Ni-NTA concentrations (5, 10, 15 mol%) (Supplementary Figs. 6-7).”

Further, to determine whether DGS-Ni-NTA lipids were required for the observed probe partitioning, we examined an alternative means of recruiting the RGG protein to membrane surfaces, which did not rely on DGS-Ni-NTA lipids. Specifically, we created a recombinant protein chimera which linked the epsin1 N-terminal homology (ENTH) domain and RGG, ENTH-RGG. The ENTH domain of Epsin1, a protein involved in endocytosis, is known to bind phosphatidylinositol-4,5-bisphosphate (PI(4,5)P2) lipids¹⁹. Therefore, to recruit ENTH-RGG to the membrane surface, we incorporated PI(4,5)P2 lipids into suspended membranes. When these membranes were incubated with ENTH-RGG, liquid-like protein condensates began to form, which fused and re-rounded upon contact. Also, the exclusion of lipid probes (BODIPY TR Ceramide) from the protein-rich regions was observed, similar to our observations upon incubation of his-RGG with suspended membranes containing DGS-Ni-NTA. The partition coefficient of the dye (K_p) was 1.24 for membranes containing PI(4,5)P2, somewhat smaller compared to the K_p value of 1.60 for the same probe in membranes containing DGS-Ni-NTA. These results suggest that exclusion of probe lipids from the protein-enriched phase occurs, regardless of the specific binding interactions used to recruit proteins to the membrane surface.

However, the degree of probe exclusion varies with the binding interaction, suggesting that the mechanism of protein-lipid interaction can play a role in the process. Moreover, transmembrane coupling of ENTH-RGG condensates was also observed when the proteins were allowed to phase separate on both sides of the membrane at the same time. These data are shown in Supplementary Figure 8, which is copied below and has been added to the Supplementary Information of our revised submission.

Supplementary Figure 8. Phase separation of ENTH-RGG protein chimera on the membrane. **a**, Schematic of recombinant protein chimera ENTH-RGG and its binding and phase separation on the membrane through interactions between the ENTH domain and PI(4,5)P2 lipids. **b**, Images showing fusion of protein-rich regions. Yellow arrowheads indicate the locations

of fusion events. Scale bars, 10 μm . **c**, Images for the case where the protein-rich phase is the continuous phase and protein-depleted phase is the dispersed phase (described as Case 3 in the main text). Scale bar, 2 μm . **d,e** Images showing coupled and uncoupled regions when proteins bind to both sides of the membrane. Scale bars, 2 μm . **f-h**, Relative intensity profile along the dotted line in the merged channel in **d** for coupled (**f**) and uncoupled (**g**) regions, and in the merged channel in **e** (**h**), where green and red lines represent the intensity from the protein and lipid channels, respectively. 1 μM of ENTH-RGG, labeled with Atto 488, was used. Membrane composition: 80 mol% DOPC, 15 mol% DOPS, 5 mol% PI(4,5)P2, and 0.5 mol% BODIPY TR-Ceramide. Buffer: 25 mM HEPES, 100 mM NaCl, pH 7.4.

The following statement was added to the main text to introduce this figure:

“Further, to determine whether DGS-Ni-NTA lipids were required for the observed probe partitioning, we examined an alternative means of recruiting the RGG protein to membrane surfaces. Specifically, we created a recombinant protein chimera which linked the epsin1 N-terminal homology (ENTH) domain and RGG, which we call ENTH-RGG. The ENTH domain is known to bind phosphatidylinositol-4,5-bisphosphate (PI(4,5)P2) lipids¹⁹. Therefore, to recruit ENTH-RGG to the membrane surface, we incorporated PI(4,5)P2 lipids into the membrane. Here, the exclusion of lipid probes (BODIPY TR Ceramide) from the protein-rich regions was also observed, with a somewhat smaller partition coefficient (K_P) of 1.24 compared to the K_P value of 1.60 for the same probe when DGS-Ni-NTA was used for recruitment of his-RGG. Transmembrane coupling of ENTH-RGG condensates was also observed (Supplementary Fig. 8). These results suggest that exclusion of probe lipids from the protein-rich phase, as well as transmembrane protein coupling, occur regardless of the specific binding interaction used to recruit proteins to the membrane surface. However, the observed variation in the extent of probe lipid exclusion suggests that protein-lipid binding interactions play a role in transbilayer coupling.”

5. Regarding the observations that different viscosities are observed in the different phases, I believe that this is an expected result given the large fraction of Ni lipids that are interacting strongly with protein. Even outside of this, the system is clearly separated into multiple surface phases, and it is expected that these phases would have different physical properties such as viscosity. Even in the absence of a large Ni lipid concentration, I would expect frictional coupling with protein would impact lipid mobility. It also follows that there would be different viscosities in the systems with three phases (protein-depleted phase, protein-enriched phase on one side, protein-enriched phases on both sides). While it is nice to see this expectation validated experimentally, it does not provide evidence for a specific coupling mechanism, as stated in the text.

Action taken: We agree that the change in recovery time during FRAP experiments could arise from a combination of effects including differences in lipid composition, differences in lipid mobility, protein-lipid interactions, etc. We have added the following sentences to the revised manuscript to suggest possible factors that could affect lipid mobility in our observations:

“The observed differences in lipid mobility could result from differences in lipid packing, but could also arise from a combination of effects including differences in lipid composition, such as enrichment of DGS-Ni-NTA lipids in the protein-rich regions, frictional interactions between proteins and lipids, and modified local bending rigidity or local spontaneous curvature due to membrane-bound proteins²⁰.”

6. I strongly disagree with the conclusion that coupling needs to be accomplished through “entropic” ordering interactions with the chains. There are many possible explanations that would involve energetic contributions to the free energy treating the membrane as an elastic sheet (as is typically done) – e.g. bending rigidity, spontaneous curvature coupling, etc. There is a ton of theory literature in this area. Of course entropy is also involved in free energy but there are many contributions to this too. I would find the results of this study to be much more unexpected if no apparent coupling were observed, since there are so many ways in which coupling could be expected to arise. This is not to say that the experimental results are not interesting, but framing them as “reducing entropy” is not particularly informative and also misleading – certainly this has not been shown here decisively.

Relevant theory has been done on a related system (minus the asymmetry present here): e.g. 10.1073/pnas.2103401118, 10.1088/1367-2630/ac320b. It seems like these (or similar) should be cited along with the past experimental studies mentioned, but may also be helpful in developing a more rigorous theoretical explanation.

Action taken: In response to the referee’s comments, we have revised our manuscript such that we move away from a specific mechanistic explanation and instead discuss possible factors that could contribute to transbilayer coupling. We present this discussion in an open ended way, acknowledging that future work will be needed to determine a specific mechanism. Additionally, we have now cited the two papers that the reviewer mentioned, along with relevant past works. The revised text (quoted above in response to introductory remarks by the same referee) reads as follows:

“It is well known that ordered lipid domains in opposing membrane leaflets frequently become coupled, or registered, by various mechanisms including line tension, membrane fluctuation, and spontaneous curvature⁸⁻¹². Transmembrane coupling of protein condensates could in principle arise from lipid domain coupling. Such coupling could be triggered by the local ordering of lipids, which could be accomplished by altering lipid packing, lipid composition, or membrane curvature, each via protein-lipid interactions (Fig. 5a). Given the diversity of factors that could contribute to our findings, future work by the community, including theoretical studies^{13,14}, will be needed to isolate the specific mechanisms responsible for transmembrane coupling of protein condensates.”

Also, we revised Figure 5a in the main text so that a specific mechanism for transmembrane coupling is not suggested. Specifically, we replaced the words related to specific mechanisms such as ‘entropy’, ‘fluctuation’, and ‘entropic domain coupling’ to ‘local ordering of the lipids by

protein condensates' and 'Transmembrane domain coupling'. Revised figure is copied below (revised figure caption in red).

Figure 5. Transmembrane coupling of protein condensates correlates with changes in lipid organization and mobility. **a**, Schematic of the **transmembrane domain coupling process**. Left: Protein condensates induce ordered lipid regions (lipids with blue heads) **with reduced conformational freedom**. Right: Protein condensates on different sides of the membrane become coupled **b,c**, Representative microscopic images from lipid channels and cartoons showing fusion events of two uncoupled regions (**b**) and two coupled regions (**c**) of membrane-protein composites over time. **d**, Aspect ratio changes over time during relaxation after fusion of two regions. Red circles indicate aspect ratio change for uncoupled regions, and blue circles for coupled regions. Dotted lines represent an exponential fit: $y(t) = A + B \cdot \exp(-t/\tau)$. **e**, Representative microscopic images in the lipid channel showing fluorescence recovery for protein-depleted, uncoupled, and coupled regions. Yellow arrows indicate photobleached regions. **f**, FRAP profile for the control (protein-free membrane, red circles), Protein-depleted (blue triangles), uncoupled (green diamonds), and coupled (black squares) regions. Shaded regions in each color represent standard deviations for each case from 5 to 17 independent measurements. **g**, Corresponding

$t_{1/2}$, time required for 50% of fluorescence recovery, from FRAP profile in **f**. Error bars indicate standard deviation. **h**, Left: Representative microscopic image in the lipid channel showing both brighter and dimmer regions for calculating partition coefficient of lipid probe. Right: Partition coefficients (K_P) as a function of different NaCl concentrations. The red bars indicate the average K_P values for each case. **i**, Schematic of our **observation that protein condensates create lipid regions with reduced lipid mobility**. Brackets in **g** and **h** show statistically significant comparisons using an unpaired, two-tailed Student's t test. * $p < 0.05$, *** $p < 0.001$, and N.S. indicates a difference that was not statistically significant. Membrane composition: 85 mol% DOPC, 15 mol% DGS-Ni-NTA with 0.5 mol% Texas Red-DHPE (**b,c,h**) or NBD-PC (**e**). 1 μ M of unlabeled his-RGG was used. Scale bars, 5 μ m.

Reviewer #3:

In Lateral compression of lipids drives transbilayer coupling of liquid-like protein condensates, Lee et al. develop methodology to investigate coupling membrane-proximal condensates located on either side of a synthetic membrane. They stretch planar membranes over an EM grid and flow an aqueous solution with phase-separating proteins (RGG, FUS) over one or both sides of the bilayer. Proteins may bind to lipids tagged Ni-DGS lipids. They demonstrate that condensates form on these membranes, – and do so at concentrations and interactions strengths far weaker than those required to phase-separate in bulk; the authors chart a phase – diagram of this membrane-condensate system.

Condensates that are allowed to interact with lipids on both sides of the membrane are found to be “coupled”: fusion-like events co-localize these transbilayer condensates. The authors propose an “entropic compression” mechanism underlying this phenomenon, whereby condensates increase the local ordering of membrane lipids, and that the entropic penalty for doing such can be abrogated by colocalized condensates in one-domain, as opposed to two. Results are demonstrated in simple phase-separating protein systems LAF-RGG and FUS.

This manuscript presents a first step at understanding how condensates interact with both sides of a lipid membrane. The methodology and results pertaining to bilayer coupling are new and interesting. I am, however, concerned with the interpretation and presentation of results pertaining to the proposed mechanism of transbilayer coupling.

Action taken: We appreciate the reviewer's careful reading and understanding of our manuscript. Below we provided a detailed response to the reviewer's concerns.

Major Comments:

1) The authors suggest an “entropic” coupling mechanism, whereby a protein condensate increases the order of the underlying lipids. This is an interesting possibility, but the claim is stated too confidently, and not adequately tested. I suggest that the authors further verify some of their

results, and temper the confidence with which they claim this mechanism in the results and discussion.

Action taken: In response to similar comments by Reviewer #2, we have modified the manuscript title to read: “**Transmembrane coupling of liquid-like protein condensates**”, which does not imply precise knowledge of the underlying mechanisms. We also revised the discussion to reflect uncertainty about the coupling mechanism. Please see our response to Reviewer #2, points 4-6. However, points unique to Reviewer #3 are addressed below.

i) This putative mechanism relies on the lipids underlying the condensate becoming more ordered, yet the authors only test that the condensates exclude disordered dye. Does an ordered-phase dye partition into the lipid phases underlying these condensates? If possible, the authors should perform experiments in the coupled system with ordered phase dye to see if this partitions into the condensate-rich phase as expected.

Action taken: This is an interesting suggestion that we also considered in the course of our work. Specifically, we already applied several dyes known to partition into liquid-ordered phases (e.g. Oregon Green-DHPE, and NBD-DHPE), as shown in Figure 4c in the main text. We observed that these dyes, along with those normally associated with disordered lipid regions (Texas Red-DHPE, BODIPY TR-Ceramide, and NBD-PC), were excluded from protein-rich regions. This result could suggest that protein condensates create an ultra-dense region within the membrane that excludes a larger variety of probe lipids than has been previously observed. However, given the broader uncertainty about the mechanism, we have now relaxed our claims about lipid compression, as noted in response to Reviewer #2, points 4-6. We have added the following sentences to the main text to note that dyes known to partition into liquid-ordered phases were also excluded from the protein-rich regions:

“Interestingly, we observed that several probe lipids known to partition into liquid-ordered phases (Oregon Green-DHPE and NBD-DHPE), along with those normally associated with liquid-disordered phases (Texas Red-DHPE, BODIPY TR-Ceramide, and NBD-PC), were excluded from the protein-rich regions. This result could suggest that protein condensates create an ultra-dense lipid region with a high degree of lipid packing that excludes a larger variety of probe lipids than has been previously observed.”

ii) Condensates could interact with the dye. One could expect that condensates could repel the dye molecule, depleting it from the phase without needing any ordering mechanism. Slower kinetics would then be related to an enrichment of protein-interacting DGS-Ni-NTA lipids in this phase. Do condensates that form in bulk solution interact with the dye at all?

Action taken: We acknowledge that interactions between protein condensates and dyes, whether attractive or repulsive, could impact the partitioning results. Our changes in response to the previous question address this point.

iii) Dyes could have differential partitioning in DOPC vs DGS. If this is the case, depletion of dyes

(and perhaps slower diffusion) within the droplets could be explained by the enrichment of DGS-Ni-NTA lipids in the domain. The authors should check that this is not the case

Action taken: First, we observed that the percentage of DGS-Ni-NTA can be lowered to 5 mol% without substantially changing the observed dye exclusion phenomena, Supplementary Figure 6 (shown below). Also, we examined the impact of DGS-Ni-NTA concentration on the extent of lipid probe exclusion by calculating the partition coefficient (K_P) of the lipid probe as a function of DGS-Ni-NTA. This analysis revealed that the partition coefficient of Texas Red-DHPE was not a function of DGS-Ni-NTA concentration, Supplementary Figure 7 (shown below). These findings suggest that lipid probe exclusion cannot be explained by enrichment of DGS-Ni-NTA in protein-rich regions of the membrane.

Supplementary Figure 6. Lipid probe exclusion from protein-rich regions with low DGS-Ni-NTA concentration in the membrane. Membrane composition: 95 mol% DOPC, 5 mol% DGS-Ni-NTA with 0.5 mol% Texas Red DHPE. Buffer: 25 mM HEPES, 100 mM NaCl, pH 7.4. 1 μM of his-RGG labeled with Atto 488 was used. Scale bars, 10 μm.

Supplementary Figure 7. Partition coefficients (K_P) of lipid probe as a function of different DGS-Ni-NTA concentrations in the membrane. Red bars indicate the average K_P values for each case. Membrane composition: 75-95 mol% DOPC, 5-25 mol% DGS-Ni-NTA with 0.5 mol% Texas Red DHPE. 1 μ M of his-RGG was used. Brackets show statistically significant comparisons using an unpaired, two-tailed Student's t test. (n.s. indicates a difference that was not statistically significant.)

The following statement was added to the main text to introduce Supplementary Figure 6 and 7:

“Meanwhile, probe lipid exclusion is unlikely to arise due to the enrichment of DGS-Ni-NTA lipids in protein-rich regions, as the extent of probe lipid partitioning remained similar for varying concentrations of DGS-Ni-NTA in the membrane (Supplementary Figs. 6-7).”

Further, to determine whether DGS-Ni-NTA lipids were required for the observed probe partitioning, we examined an alternative means of recruiting the RGG protein to membrane surfaces, which did not rely on DGS-Ni-NTA lipids. Specifically, we created a recombinant protein chimera which linked the epsin1 N-terminal homology (ENTH) domain and RGG, ENTH-RGG. The ENTH domain of Epsin1, a protein involved in endocytosis, is known to bind phosphatidylinositol-4,5-bisphosphate (PI(4,5)P2) lipids¹⁹. Therefore, to recruit ENTH-RGG to the membrane surface, we incorporated PI(4,5)P2 lipids into suspended membranes. When these membranes were incubated with ENTH-RGG, liquid-like protein condensates began to form, which fused and re-rounded upon contact. Also, the exclusion of lipid probes (BODIPY TR Ceramide) from the protein-rich regions was observed, similar to our observations upon incubation of his-RGG with suspended membranes containing DGS-Ni-NTA. The partition coefficient of the dye (K_P) was 1.24 for membranes containing PI(4,5)P2, somewhat smaller compared to the K_P value of 1.60 for the same probe in membranes containing DGS-Ni-NTA. These results suggest that exclusion of probe lipids from the protein-enriched phase occurs, regardless of the specific binding interactions used to recruit proteins to the membrane surface. However, the degree of probe exclusion varies with the binding interaction, suggesting that the mechanism of protein-lipid interaction can play a role in the process. Moreover, transmembrane coupling of ENTH-RGG condensates was also observed when the proteins were allowed to phase separate on both sides of the membrane at the same time. These data are shown in Supplementary Figure 8, which is copied below and has been added to the Supplementary Information of our revised submission.

Supplementary Figure 8. Phase separation of ENTH-RGG protein chimera on the membrane. **a**, Schematic of recombinant protein chimera ENTH-RGG and its binding and phase separation on the membrane through interactions between the ENTH domain and PI(4,5)P2 lipids. **b**, Images showing fusion of protein-rich regions. Yellow arrowheads indicate the locations of fusion events. Scale bars, 10 μm . **c**, Images for the case where the protein-rich phase is the continuous phase and protein-depleted phase is the dispersed phase (described as Case 3 in the main text). Scale bar, 2 μm . **d,e** Images showing coupled and uncoupled regions when proteins bind to both sides of the membrane. Scale bars, 2 μm . **f-h**, Relative intensity profile along the dotted line in the merged channel in **d** for coupled (**f**) and uncoupled (**g**) regions, and in the merged channel in **e** (**h**), where green and red lines represent the intensity from the protein and lipid channels, respectively. 1 μM of ENTH-RGG, labeled with Atto 488, was used. Membrane

composition: 80 mol% DOPC, 15 mol% DOPS, 5 mol% PI(4,5)P2, and 0.5 mol% BODIPY TR-Ceramide. Buffer: 25 mM HEPES, 100 mM NaCl, pH 7.4.

The following statement was added to the main text to introduce Supplementary Figure 8:

“Further, to determine whether DGS-Ni-NTA lipids were required for the observed probe partitioning, we examined an alternative means of recruiting the RGG protein to membrane surfaces. Specifically, we created a recombinant protein chimera which linked the epsin1 N-terminal homology (ENTH) domain and RGG, which we call ENTH-RGG. The ENTH domain is known to bind phosphatidylinositol-4,5-bisphosphate (PI(4,5)P2) lipids¹⁹. Therefore, to recruit ENTH-RGG to the membrane surface, we incorporated PI(4,5)P2 lipids into the membrane. Here, the exclusion of lipid probes (BODIPY TR Ceramide) from the protein-rich regions was also observed, with a somewhat smaller partition coefficient (K_P) of 1.24 compared to the K_P value of 1.60 for the same probe when DGS-Ni-NTA was used for recruitment of his-RGG. Transmembrane coupling of ENTH-RGG condensates was also observed (Supplementary Fig. 8). These results suggest that exclusion of probe lipids from the protein-rich phase, as well as transmembrane protein coupling, occur regardless of the specific binding interaction used to recruit proteins to the membrane surface. However, the observed variation in the extent of probe lipid exclusion suggests that protein-lipid binding interactions play a role in transbilayer coupling.”

iv) This mechanism implies that a membrane composed of a lipid that is more ordered would present less of an effect, as there is less entropy to be “lost”. Can the authors perform this experiment in a membrane composed of ordered-phase lipids, and see if condensates still couple across the bilayer?

Action taken: We appreciate the reviewer’s constructive suggestion. We agree that the proposed experiment would be a good way to test our hypothesis. Unfortunately, ordered-phase lipids generally have melting temperatures higher than room temperature, such that the resulting membranes are very brittle at room temperature and prone to rupturing in our suspended bilayer system. Because we are unable to perform a definitive experiment to identify the coupling mechanism, we have revised the manuscript to provide multiple possible mechanisms for transmembrane coupling rather than suggesting one particular mechanism. The modified text is quoted below:

“It is well known that ordered lipid domains in opposing membrane leaflets frequently become coupled, or registered, by various mechanisms including line tension, membrane fluctuation, and spontaneous curvature⁸⁻¹². Transmembrane coupling of protein condensates could in principle arise from lipid domain coupling. Such coupling could be triggered by the local ordering of lipids, which could be accomplished by altering lipid packing, lipid composition, or membrane curvature, each via protein-lipid interactions (Fig. 5a). Given the diversity of factors that could contribute to our findings, future work by the community, including theoretical studies^{13,14}, will be needed to isolate the specific mechanisms responsible for transmembrane coupling of protein condensates.”

2) The coupled-bilayer experiments suggest coexistence of three distinct quasi-2D liquid phases, – protein rich on both sides, protein rich on one side, and protein poor. This is reminiscent of three-phase coexistence in asymmetric, multicomponent lipid membranes (vide Collins & Keller 2008) and could, in spirit, be the same phenomena.

i) Can the authors comment on whether this is a correct interpretation? If so, why avoid calling this phenomenon three-phase coexistence? If not, why?

Action taken: We appreciate the reviewer’s comment, and we believe that the coexistence of three phases in asymmetric, multicomponent lipid membranes reported in the paper the reviewer mentioned (M. D. Collins and S. L. Keller *Proc. Natl. Acad. Sci. U. S. A.* **105**, 124–128, 2008) would be related to our observation of transmembrane coupling of protein condensates. Please refer to the similar comment in the introductory remark of Reviewer #2. We have added the following sentence to the revised manuscript to point out this similarity, citing the above-mentioned paper (M. D. Collins and S. L. Keller *Proc. Natl. Acad. Sci. U. S. A.* **105**, 124–128, 2008):

“Note that this observation can also be described as the coexistence of three different phases⁸.”

ii) asymmetric membranes grant a bilayer some coupling across leaflets, as can curvature and membrane spanning proteins. Can the authors comment on the magnitude of these effects compared to their proposed mechanism? In the Discussion the authors calculate the order of these effects, – would this be a small correction to the other coupling mechanisms, or a dominant effect? Real cells often have different condensates on different sides of the membrane presenting differences in possibly the scale and the sign of these effects. The authors should temper their claims about transbilayer communication via this mechanism alone.

Action taken: We acknowledge that multiple mechanisms can drive coupling in cells, such as transmembrane proteins, curvature, channels, etc., and that the coupling we observed is unlikely to act alone, given this complexity. We appreciate the reviewer’s very interesting question about the relative magnitudes of these effects. These are difficult to calculate at present given the nascent understanding of the observed coupling mechanism and the contexts in which it would interact with the other mechanisms. We believe that it will be a very interesting area for future work. We have revised our manuscript to reflect all of the above comments as follows (newly added text in red):

“From a biological perspective, it is increasingly clear that liquid-like proteins help to organize critical structures and events at biological membranes, from assembly of cell-cell junctions to the budding of trafficking vesicles^{21–23}. Importantly, each of these assemblies involves protein-protein interactions on both surfaces of the membrane. In such processes, we speculate that the **transbilayer coupling of protein condensates observed here** works in concert **with other factors including** transmembrane proteins, lipid phase separation²⁴, **and membrane curvature** to achieve robust transbilayer communication.

Determining the relative magnitude of each effect, while presently unknown, is an interesting subject for future work.”

iii) Is the membrane symmetric in DGS-Ni-NTA concentration? If so, the domains should (after a very long time) all be coupled across the membrane. Should I interpret the effects as a kinetic effect but indicative of equilibrium phase behavior?

Action taken: We expect the concentration of DGS-Ni-NTA lipids to be the same in both leaflets of the membrane, as the lipids that comprise each leaflet are supplied from the same oil bath, into which all lipids were originally dissolved. The degree of coupling between protein-rich phases increases over time in our experiments until, finally reaching a completely coupled state in many cases. Therefore, we agree with the reviewer’s assessment that full coupling likely represents the equilibrium state. We have added the following sentence in the revised manuscript to clarify the point:

“Notably, we observed more domain coupling over time, which likely represents the approach of the system to equilibrium.”

3) In Figure 2 the authors claim that bilayer coupling increases the critical temperature of membrane-bound condensates. I find the presentation of this claim troubling, as it conflates critical temperatures with transition temperature. The mixtures do not show critical-like fluctuations, and are likely not near a critical point.

i) The authors should clarify the language involved in this statement (e.g change “critical temperature” to “transition temperature”) to be less suggestive of being near a critical point.

Action taken: We appreciate this important comment from the reviewer. In response to the same comment by Reviewer #2, we have changed the wording “critical temperature” to “transition temperature”.

ii) Another concern is that this conflates surface behavior with bulk behavior, when the transition on the membrane could be a qualitatively different transition (vide Rouches et al PNAS 2021, Zhao et al NJP 2021). The authors should consider whether they are changing the bulk phase boundary, the surface phase boundary, or both.

Action taken: We have used 1 μM of RGG protein for our experiments, which is at least one order of magnitude below the concentration at which RGG proteins phase separate in solution (B. Schuster et al. Nat. Comm. 2018). Therefore, protein phase separation is only expected to occur on the membrane in our experiments, in accordance with our observations. We make this point in the main text as follows (newly added text in red):

“Further, RGG is highly soluble in aqueous buffers, such that phase separation in solution only occurs at concentrations greater than 10 μM ^{18,25}, substantially above the concentration used in our experiments with membranes. In this way, we could clearly

differentiate protein phase separation on membranes from phase separation in the surrounding solution, **which was not observed under our experimental conditions.**"

4) Related: The phase-diagram the authors draw in Figure 2 is not in terms of a conserved quantity. The only conserved quantity here is the DGS-Ni-NTA concentration. Is there a way to measure the DGS concentration in each phase? If so the authors should draw a phase-diagram in this space, which would be more physically interpretable.

i) related: Figures 1G,H are nearly a phase diagram in the correct coordinates, traversing across a tie-line but chop the space up into various cases corresponding to surface-dilute (case 1), surface-dense (case 4), surface-coexistence (case 2 & 3). Why divide the space up into these categories and not measure the area-fraction of the two phases?

Action taken: The main message we want to present to the reader with Figures 1G and H is that the fraction of the membrane area covered by the protein-rich phase increases with increasing Ni-NTA concentration and decreasing NaCl concentration. This is the behavior expected for phase separation of the RGG domain in solution¹⁸, and we are simply confirming that this behavior is maintained when RGG binds to the membrane surface.

However, in response to the referee's concern and to make these data more informative for the reader, we have calculated the fraction of the membrane surface covered by the protein-rich phase at each combination of DGS-Ni-NTA and NaCl, Supplementary Figure 2 (below). Here we see that the area fraction of the protein-rich phase, while somewhat heterogeneous, generally increases with increasing DGS-Ni-NTA concentration and NaCl concentration.

Supplementary Figure 2. Area fraction of protein-rich regions as a function of NaCl concentration and DGS-Ni-NTA concentration. Total 20 lipid membranes were analyzed from 2-3 independent experiments for each condition. Bars represent the average area fraction for

each case. Membrane composition: 75-95 mol% DOPC, 5-25 mol% DGS-Ni-NTA. 1 μ M of his-RGG labeled with Atto 488 were used.

The following statement was added to the main text to introduce this figure:

“Similarly, the fraction of the membrane area covered by the protein-rich phase increased with decreasing NaCl concentration and increasing Ni-NTA concentration (Supplementary Fig. 2).”

5) The authors cite that Laf-RGG phase separates in 3D at near 10 micromolar concentration. The cited paper (Elbaum-Garfinkle 2015) shows that, depending on salt concentration, this can be reduced to \sim one micromolar. Can the authors present their “phase diagram” of a unary LAF-RGG solution, perhaps in the supplement?

Action taken: In the paper we cited Elbaum-Garfinkle et al. PNAS 2015, in which it was shown that the concentration of full-length LAF-1 protein required for phase separation in solution can be reduced to \sim 1 μ M depending on NaCl concentration¹⁸, as the referee states. However, the same paper demonstrated that the isolated single RGG domain, which is the portion of LAF-1 protein used in our study did not phase separate until it reached a concentration of approximately 20 μ M at 20 °C with 125 mM NaCl. Similarly, another paper (B. Schuster et al. Nat. Commun. 9, 2018) reported that 12 μ M of the isolated single RGG domain was required for phase separation, even when the temperature was reduced to 15 °C with 150 mM NaCl²⁵. We have now cited both papers in the revised manuscript to make it clearer that the isolated RGG domain would not be expected to phase separate in solution under our experimental conditions.

Minor comments:

1) The authors say that they “increase the strength of protein membrane interactions” by increasing the amount of DGS-Ni-NTA lipid in the membrane. This seems more like they increase the number of these interactions, while holding their strength fixed. Can the authors clarify this language?

Action taken: We have changed the wording, “the strength of protein-membrane interactions” to “the density of protein-membrane interactions” to reflect the referee’s comment.

2) The authors write “... the cooperative relationship between protein binding and phase separation...”: It is very unclear what they mean here.

Action taken: We have changed the wording as follows (revised text in red):

“The observation of multiple cases for the same condition did not arise from variation in lipid composition between the grid holes, as a non-phase separating protein, green fluorescent protein (GFP) bound each hole approximately equally (Supplementary Fig. 3). However, it may arise from imperfect mixing upon addition of protein to solution, leading to variation in the extent of protein binding across the grid.”

3) The authors use the term “simultaneously” to mean “co-localized”, or something else. Do the domains really form at the same time?

Action taken: We used the term “simultaneously” to refer to protein phase separation, which occurred on both surfaces of the membrane at the same time. We have changed the wording ‘simultaneously’ to ‘at the same time’ in the main text to clarify the meaning.

4) An implicit claim behind the authors interpretation is that, although the condensates “communicate” with one another, there is no additional stabilization of condensates across the bilayer. Is it possible that this coupling can stabilize condensates in regimes where they don’t form on membranes with only one accessible bilayer? I would guess this would be a small regime, and maybe not accessible.

Action taken: This is an interesting question that may have biological relevance. We plan to investigate this concept in our future work.

5) On the second paragraph of results and discussion, the authors write “Further RGG is high soluble” should read “highly soluble”.

Action taken: Typo was fixed (from ‘high’ to ‘highly’).

References

1. Yuan, F. *et al.* Membrane bending by protein phase separation. *Proc. Natl. Acad. Sci. U. S. A.* **118**, e2017435118 (2021).
2. Perdikari, T. M. *et al.* A predictive coarse-grained model for position-specific effects of post-translational modifications. *Biophys. J.* **120**, 1187–1197 (2021).
3. Baumgart, T., Hunt, G., Farkas, E. R., Webb, W. W. & Feigenson, G. W. Fluorescence probe partitioning between Lo/Ld phases in lipid membranes. *Biochim. Biophys. Acta - Biomembr.* **1768**, 2182–2194 (2007).
4. Klymchenko, A. S. & Kreder, R. Fluorescent probes for lipid rafts: From model membranes to living cells. *Chem. Biol.* **21**, 97–113 (2014).
5. Yim, H. *et al.* Rearrangement of lipid ordered phases upon protein adsorption due to multiple site binding. *Phys. Rev. Lett.* **96**, 198101 (2006).
6. Mangiarotti, A. & Bagatolli, L. A. Impact of macromolecular crowding on the mesomorphic behavior of lipid self-assemblies. *Biochim. Biophys. Acta - Biomembr.* **1863**, 183728 (2021).
7. Beltramo, P. J., Van Hooghten, R. & Vermant, J. Millimeter-area, free standing, phospholipid bilayers. *Soft Matter* **12**, 4324–4331 (2016).
8. Collins, M. D. & Keller, S. L. Tuning lipid mixtures to induce or suppress domain formation across leaflets of unsupported asymmetric bilayers. *Proc. Natl. Acad. Sci. U. S. A.* **105**, 124–128 (2008).
9. Blosser, M. C., Honerkamp-Smith, A. R., Han, T., Haataja, M. & Keller, S. L. Transbilayer Colocalization of Lipid Domains Explained via Measurement of Strong Coupling Parameters. *Biophys. J.* **109**, 2317–2327 (2015).
10. Haataja, M. P. Lipid Domain Co-localization Induced by Membrane Undulations. *Biophys. J.* **112**, 655–662 (2017).
11. Galimzyanov, T. R., Kuzmin, P. I., Pohl, P. & Akimov, S. A. Undulations Drive Domain Registration from the Two Membrane Leaflets. *Biophys. J.* **112**, 339–345 (2017).
12. Sarmiento, M. J., Hof, M. & Šachl, R. Interleaflet Coupling of Lipid Nanodomains – Insights From in vitro Systems. *Front. Cell Dev. Biol.* **8**, 284 (2020).
13. Rouches, M., Veatch, S. L. & Machta, B. B. Surface densities prewet a near-critical membrane. *Proc. Natl. Acad. Sci. U. S. A.* **118**, e2103401118 (2021).
14. Zhao, X., Bartolucci, G., Honigmann, A., Jülicher, F. & Weber, C. A. Thermodynamics of wetting, prewetting and surface phase transitions with surface binding. *New J. Phys.* **23**, 123003 (2021).
15. Leptihn, S. *et al.* Constructing droplet interface bilayers from the contact of aqueous droplets in oil. *Nat. Protoc.* **8**, 1048–1057 (2013).

16. Lee, Y., Lee, H.-R., Kim, K. & Choi, S. Q. Static and Dynamic Permeability Assay for Hydrophilic Small Molecules Using a Planar Droplet Interface Bilayer. *Anal. Chem.* **90**, 1660–1667 (2018).
17. Beltramo, P. J., Scheidegger, L. & Vermant, J. Toward Realistic Large-Area Cell Membrane Mimics: Excluding Oil, Controlling Composition, and Including Ion Channels. *Langmuir* **34**, 5880–5888 (2018).
18. Elbaum-Garfinkle, S. *et al.* The disordered P granule protein LAF-1 drives phase separation into droplets with tunable viscosity and dynamics. *Proc. Natl. Acad. Sci. U. S. A.* **112**, 7189–7194 (2015).
19. Ford, M. G. J. *et al.* Curvature of clathrin-coated pits driven by epsin. *Nature* **419**, 361–366 (2002).
20. Zhang, L. & Granick, S. Slaved diffusion in phospholipid bilayers. *Proc. Natl. Acad. Sci. U. S. A.* **102**, 9118–9121 (2005).
21. Beutel, O., Maraspini, R., Pombo-García, K., Martin-Lemaitre, C. & Honigmann, A. Phase Separation of Zonula Occludens Proteins Drives Formation of Tight Junctions. *Cell* **179**, 923–936 (2019).
22. Schwayer, C. *et al.* Mechanosensation of Tight Junctions Depends on ZO-1 Phase Separation and Flow. *Cell* **179**, 937–952 (2019).
23. Day, K. J. *et al.* Liquid-like protein interactions catalyse assembly of endocytic vesicles. *Nat. Cell Biol.* **23**, 366–376 (2021).
24. Wang, H.-Y. *et al.* Coupling of protein condensates to ordered lipid domains determines functional membrane organization. *Sci. Adv.* **9**, eadf6205 (2023).
25. Schuster, B. S. *et al.* Controllable protein phase separation and modular recruitment to form responsive membraneless organelles. *Nat. Commun.* **9**, 2985 (2018).

REVIEWER COMMENTS

Reviewer #1 (Remarks to the Author):

The authors have revised the manuscript thoroughly according to reviewers' comments.

Reviewer #2 (Remarks to the Author):

Major:

I appreciate that the authors have altered the discussions of mechanism and have edited the title. These changes greatly improve the revised manuscript.

This said, I maintain that both major findings that contribute to the authors statement concluding that "protein condensates locally order membrane lipids" (quote from abstract) is most likely a result of the lipids occupying these domains being largely made up of the coupling lipids, either the Ni lipids used most frequently, or the PIP2 lipids used in Sup Fig 8. This would lead to the trivial depletion of the labeled lipid probe and the slowing of lipid mobility, and not some more fundamental biophysical mechanism of "ordering," which is implied. I do not mean to state that a description of this trivial mechanism makes the work unpublishable, although I do think it necessarily would lead to a shorter manuscript with less speculation.

This conclusion is supported by the new data presented in this revision. To spell this out: the authors state that when 5% or 10% of the Ni lipid is included, then the protein domains occupy 5% or 10% of the membrane surface area. Since the Ni lipid strongly interacts with these proteins, this observation suggests that, at least under these conditions, that membranes in the leaflet adjacent to protein rich domains are nearly exclusively occupied by Ni lipids. If this were to be the case, one would expect to observe exclusion of a lipid probe from these domains, even if the probe molecule has equal access to available sites compared to other bulk lipid. In the extreme limit that domains are 100% occupied by Ni lipid, with no available sites for uncoupled lipids in 1 leaflet, then you would expect a partition coefficient of $K_p = (1+1)/(0+1) = 2$. This is the number reported in Sup fig 7 for headgroup labeled TxRed. Odds are good that these domains are not 100% Ni lipid, but it seems reasonable to think that they are at least 80%, which would get you closer to $K_p = (1+1)/(.2+1) = 1.67$, which is close to what is reported to be observed for a chain labeled probe. My best guess is that the headgroup labeled probes experience extra exclusionary forces due to their bulk, rigidity and/or charge, as is acknowledged by the authors. For the case of the PIP2 coupling (in Sup Fig 8), the authors state that the K_p for the chain labeled probe drops to 1.24 – maybe in this case the PIP lipid occupies 40% of the lipids within protein rich domains, which would trivially give rise to this partition coefficient: $(1+1)/(1+.6) = 1.25$.

All of this is to say that the reason for the probe exclusion is most likely trivial – due to their being many, many fewer sites available to the probe lipid. This is not how this result is presented– instead there is a long speculation about more complicated (and more interesting) possibilities without mention of this trivial alternative. These speculations are only appropriate if there are effects beyond the trivial one described here. For example, maybe $\langle K_p \rangle$ seeming to be slightly larger than 2 in Sup fig 7 suggests some subtle depletion of the probe from the opposing leaflet?

This framing is also relevant to figure 5, where the authors draw conclusions from an observation that probe depletion is reduced when NaCl concentration increases, but they also show that this same treatment leads to an increase in the area fraction occupied by domains (Sup fig .2) Together, it seems reasonable to conclude that the concentration of Ni lipids within protein domains is reduced (because the same number spread out over a larger area), allowing there to be more sites for uncoupled lipids, including lipid probes. This would be a simple explanation for the change in probe partitioning reported.

Also in Figure 5, the authors report slowed mobility of the probe. Here it would make sense to compare to an expected result in which 80% (or similar %) of lipids in the leaflet adjacent to the domain are immobilized. I would expect this would slow diffusion of a free probe to some extent, possibly enough to be consistent with the subtle slowing reported in figure 5.

Minor:

In the abstract, both biological examples of liquid-like condensates given involve transmembrane proteins, where trans bilayer coupling is trivial. Is there a better biological motivation that does not include transmembrane proteins?

I recommend not using the word "ordering" until it is defined. In the membrane field, the word "ordering" often refers to an increase in chain order parameters, which is not explored here. I am not sure, but I think here it is used to refer to trapping of certain lipids within domains?

Conventionally, the word "coupling" means that there is a cross-term in the free energy that involves thermodynamic parameters associated with each leaflet of the bilayer. In the text, this word is used regularly to describe times when domains come into registration, which is a consequence of a certain type of coupling but not coupling itself. I recommend changing how this word is used throughout to adhere to its conventional definition. Alternatively, the word should be re-redefined somewhere early in the text.

For Sup fig 8, I am guessing that C/D represent different conditions of some sort? Only one condition is stated in the figure caption.

Reviewer #3 (Remarks to the Author):

The authors have returned a strengthened manuscript that addresses my comments, and many of the issues raised by all three reviewers. I have one minor comment that must be addressed before I'd be comfortable

1) I am still troubled by the presentation of Figure 2 and the data on salt concentration, specifically:
– lines 166-168: "... when the transition temperature is reached, the two phases have the same concentration and therefore are indistinguishable from one another, such that one homogeneous phase exists.". The only location where two phases become indistinguishable from one another is the critical point. Generally, for a first-order transition, there is a miscibility gap between the two phases at the transition point. I suggest that the authors remove this sentence (as it adds little to the text and is incorrect), or correct it.

Response to Reviews for the submission NCOMMS-23-09926A

Title

Transmembrane coupling of liquid-like protein condensates

Authors

Yohan Lee, Sujin Park, Feng Yuan, Carl C. Hayden, Liping Wang, Eileen M. Lafer, Siyoung Q. Choi and Jeanne C. Stachowiak

We thank the reviewers for thoroughly reviewing our work and providing helpful comments and suggestions. Below we respond to each of the reviewers' comments and describe the actions we have taken to address them. The changes we have made to the text are highlighted in the revised manuscript. You will see below that, *while in a few cases we could not fully understand how the reviewer interpreted our data, we have made our best effort to alter the manuscript in ways that acknowledge and address the spirit of their concerns*. We hope the reviewers agree that we have significantly strengthened the manuscript by responding to their comments.

Reviewer Comments:

Reviewer #1:

The authors have revised the manuscript thoroughly according to reviewers' comments.

Reviewer #2:

Major comments

1) I appreciate that the authors have altered the discussions of mechanism and have edited the title. These changes greatly improve the revised manuscript.

This said, I maintain that both major findings that contribute to the authors' statement concluding that "protein condensates locally order membrane lipids" (quote from abstract) is most likely a result of the lipids occupying these domains being largely made up of the coupling lipids, either the Ni lipids used most frequently, or the PIP2 lipids used in Sup Fig 8. This would lead to the trivial depletion of the labeled lipid probe and the slowing of lipid mobility, and not some more fundamental biophysical mechanism of "ordering," which is implied. I do not mean to state that a description of this trivial mechanism makes the work unpublishable, although I do think it necessarily would lead to a shorter manuscript with less speculation.

This conclusion is supported by the new data presented in this revision. To spell this out: the authors state that when 5% or 10% of the Ni lipid is included, then the protein domains occupy 5% or 10% of the membrane surface area. Since the Ni lipid strongly interacts with these proteins, this observation suggests that, at least under these conditions, that membranes in the leaflet adjacent to protein rich domains are nearly exclusively occupied by Ni lipids. If this were to be the case, one would expect to observe exclusion of a lipid probe from these domains, even if the probe molecule has equal access to available sites compared to other bulk lipids.

Action taken: We appreciate the reviewer's careful reading of our manuscript. First, we fully acknowledge the reviewer's comment that enrichment of DGS-Ni-NTA (or PI(4,5)P2) lipids in the protein-rich regions could result in the exclusion of probe lipids from protein-rich regions. We apologize for the misleading statement that probe lipid exclusion is unlikely to arise from the enrichment of DGS-Ni-NTA lipids in protein-rich regions from Supplementary Figure 7 (copied below). We initially thought the enrichment of DGS-Ni-NTA lipids in protein-rich regions might not contribute to the probe lipid exclusion simply based on the similar partition coefficients (K_P) for different DGS-Ni-NTA concentrations in Supplementary Figure 7. However, as noted by the reviewer, Supplementary Figure 2 (copied below), shows that the area fraction of protein-rich regions increases with increasing DGS-Ni-NTA concentration in the membrane. One possible explanation for these two findings is that the protein-rich region becomes enriched to some degree in DGS-Ni-NTA lipids, as the referee suggests. Once a stable degree of enrichment is achieved within these regions, increasing the DGS-Ni-NTA concentration serves only to increase the area fraction of the protein-rich phase.

Supplementary Figure 7. Partition coefficients (K_P) of lipid probe as a function of different DGS-Ni-NTA concentrations in the membrane. Red bars indicate the average K_P values for each case. Membrane composition: 75-95 mol% DOPC, 5-25 mol% DGS-Ni-NTA with 0.5 mol% Texas Red DHPE. 1 μ M of his-RGG was used. Brackets show statistically significant comparisons using an unpaired, two-tailed Student's t test. (n.s. indicates a difference that was not statistically significant.)

Supplementary Figure 2. Area fraction of protein-rich regions as a function of NaCl concentration and DGS-Ni-NTA concentration. Total 20 lipid membranes were analyzed from 2-3 independent experiments for each condition. Bars represent the average area fraction for each case. Membrane composition: 75-95 mol% DOPC, 5-25 mol% DGS-Ni-NTA. 1 μ M of his-RGG labeled with Atto 488 was used.

In response to the reviewer's comment, we have altered the text as follows (newly added text in red):

“Why are diverse probe lipids excluded from the protein-enriched phase? Probe exclusion arises due to the local reorganization of lipids, which could encompass changes in lipid packing¹, lipid composition (enrichment of DGS-Ni-NTA lipids in protein-rich regions), or the degree of lipid hydration owing to the presence of proteins², each via protein-lipid interactions. To quantify the extent of probe lipid exclusion, we defined a partition coefficient (K_P) as $K_P = I_B / I_D$, where I_B and I_D indicate the fluorescence intensity of the brighter and the dimmer regions in the lipid channel after subtracting the background intensity, respectively³. Based on the definition of K_P , the more the probes are excluded from the dimmer region, the higher the K_P value will become. For example, the K_P value can vary from one (in the case of no exclusion) to an arbitrarily high value (in the case of very strong exclusion).

Interestingly, the extent of probe lipid (Texas Red-DHPE) exclusion remained constant across a range of DGS-Ni-NTA concentrations (5, 15, 25 mol%) (Supplementary Figs. 6-7). Meanwhile, the area fraction of protein-rich regions increased over the same range of DGS-Ni-NTA concentrations (Supplementary Fig. 2). One possible explanation for these two findings is that the protein-rich region becomes enriched to some degree in DGS-Ni-NTA lipids. Once a stable degree of enrichment is achieved within these regions, increasing the DGS-Ni-NTA concentration serves only to increase the area fraction of the protein-rich phase.”

Also, we acknowledge that the term ‘ordering’ we used could be confusing, as some readers will attach a specific thermodynamic meaning to it. Therefore, we have replaced the term ‘ordering’ with the more general term, ‘reorganization’, throughout the paper so that a specific mechanism is not implied. As such, we have revised the statement in the abstract “...suggest that protein condensates locally order membrane lipids.” as follows (newly added text in red):

“...suggest that protein condensates locally **reorganize** membrane lipids.”

In addition, we revised Figure 5a in the main text such that the terms ‘ordered lipids’ and ‘local ordering’ were replaced with ‘reorganized lipids’ and ‘local reorganization’. The revised figure is copied below (revised figure caption in red).

Figure 5. Transmembrane coupling of protein condensates correlates with changes in lipid organization and mobility **a**, Schematic of the transmembrane coupling process. Left: Protein condensates induce local reorganization of lipids (lipids with blue heads). Right: Protein condensates on different sides of the membrane become coupled **b,c**, Representative microscopic images from lipid channels and cartoons showing fusion events of two uncoupled regions (**b**) and two coupled regions (**c**) of membrane-protein composites over time. **d**, The

Aspect ratio changes over time during relaxation after the fusion of two regions. Red circles indicate aspect ratio change for uncoupled regions, and blue circles for coupled regions. Dotted lines represent an exponential fit: $y(t) = A + B \cdot \exp(-t/\tau)$. **e**, Representative microscopic images in the lipid channel showing fluorescence recovery for protein-depleted, uncoupled, and coupled regions. Yellow arrows indicate photobleached regions. **f**, FRAP profile for the control (protein-free membrane, red circles), Protein-depleted (blue triangles), uncoupled (green diamonds), and coupled (black squares) regions. Shaded regions in each color represent standard deviations for each case from 5 to 17 independent measurements. **g**, Corresponding $t_{1/2}$, time required for 50% of fluorescence recovery, from FRAP profile in **f**. Error bars indicate standard deviation. **h**, Left: Representative microscopic image in the lipid channel showing both brighter and dimmer regions for calculating the partition coefficient of the lipid probe. Right: Partition coefficients (K_P) as a function of different NaCl concentrations. The red bars indicate the average K_P values for each case. **i**, Schematic of our observation that protein condensates create lipid regions with reduced lipid mobility. Brackets in **g** and **h** show statistically significant comparisons using an unpaired, two-tailed Student's t test. * $p < 0.05$, *** $p < 0.001$, and N.S. indicates a difference that was not statistically significant. Membrane composition: 85 mol% DOPC, 15 mol% DGS-Ni-NTA with 0.5 mol% Texas Red-DHPE (**b,c,h**) or NBD-PC (**e**). 1 μM of unlabeled his-RGG was used. Scale bars, 5 μm .

Accordingly, we have removed the following statements:

“Notably, the term ordered lipids does not refer to phase-separated lipid regions composed of saturated lipids and sterols, as reported elsewhere in the literature. Instead, we use the term ‘lipid ordering’ to indicate that the conformational freedom of the lipids is locally reduced due to the presence of protein condensates.”

We also revised the phrase in Discussion section as follows (newly added text in red):

“In particular, we demonstrate that a protein condensate on one side of the membrane can be detected by a condensate on the other side of the membrane through their mutual influence on the reorganization of the underlying lipids, a process that does not require a discontinuity in lipid composition, or direct contact between proteins on the two surfaces of the membrane.”

2) In the extreme limit that domains are 100% occupied by Ni lipid, with no available sites for uncoupled lipids in 1 leaflet, then you would expect a partition coefficient of $K_p = (1+1)/(0+1) = 2$. This is the number reported in Sup fig 7 for a headgroup labeled TxRed. Odds are good that these domains are not 100% Ni lipid, but it seems reasonable to think that they are at least 80%, which would get you closer to $K_p = (1+1)/(.2+1) = 1.67$, which is close to what is reported to be observed for a chain labeled probe. My best guess is that the headgroup labeled probes experience extra exclusionary forces due to their bulk, rigidity and/or charge, as is acknowledged by the authors.

For the case of the PIP2 coupling (in Sup Fig 8), the authors state that the K_P for the chain labeled probe drops to 1.24 – maybe in this case the PIP lipid occupies 40% of the lipids within protein rich domains, which would trivially give rise to this partition coefficient: $(1+1)/(1+.6) = 1.25$.

Author response: We appreciate the reviewer taking the time to think carefully about our data. We have spent considerable time trying to understand the reviewer's comment here. Unfortunately, we could not figure out how the reviewer calculated the partition coefficients (K_P). Specifically, the reviewer used the formula $K_P = (1+1)/(0+1) = 2$ for the case in which the domains were 100% occupied by Ni-NTA lipids with no available sites for probe lipids. We could not understand the reason for the addition of the values of 1 in both the numerator and the denominator. This is not how we calculate the partition coefficient in our work, as explained below. We regret that our manuscript seems to have caused confusion for the reviewer.

We calculated partition coefficients according to the equation, $K_P = I_B / I_D$, where I_B and I_D indicate the fluorescence intensity of the brighter and dimmer regions in the lipid channel, respectively. Both I_B and I_D are obtained after subtracting the background fluorescence intensity. Based on this definition, the partition coefficient would become very large (certainly greater than 2) if all probe lipids were excluded due to 100% of Ni-NTA in the protein-rich regions.

We regret that we were not able to fully understand the comments by the reviewer and hope that they will understand that we are making our best effort to incorporate the spirit of their comments in the changes we have made to the manuscript, as described below.

To further clarify the text, we have modified our description of the partition coefficient calculation to emphasize that it can vary from one (in the case of no depletion) to arbitrarily high (in the case of very strong depletion) as follows (newly added text in red):

“Based on the definition of K_P , the more the probes are excluded from the dimmer region, the higher the K_P value will become. For example, the K_P value can vary from one (in the case of no exclusion) to an arbitrarily high value (in the case of very strong exclusion).”

3) All of this is to say that the reason for the probe exclusion is most likely trivial – due to their being many, many fewer sites available to the probe lipid. This is not how this result is presented – instead there is a long speculation about more complicated (and more interesting) possibilities without mention of this trivial alternative. These speculations are only appropriate if there are effects beyond the trivial one described here. For example, maybe $\langle K_P \rangle$ seeming to be slightly larger than 2 in Sup fig 7 suggests some subtle depletion of the probe from the opposing leaflet?

Action taken: As the reviewer suggested, we have **removed** the following long speculation in the revised manuscript (removed text in blue / newly added text in red):

“Notably, the conjugation of a fluorophore to a lipid substantially increases its molecular weight, creating a bulky amphiphile that often disrupts the packing of other membrane lipids. As a result, most probe lipids are known to be excluded from liquid-ordered and

solid-like membrane phases. This reasoning suggests that altered lipid packing due to assembly of protein condensates on membrane surfaces could explain probe lipid exclusion, similar to the previously reported rearrangement of lipid packing owing to the adsorption of proteins with multiple binding sites. Interestingly, we observed that several probe lipids known to partition into liquid-ordered phases (Oregon Green-DHPE and NBD-DHPE), along with those normally associated with liquid-disordered phases (Texas Red-DHPE, BODIPY TR-Ceramide, and NBD-PC), were excluded from the protein-rich regions. This result could suggest that protein condensates create an ultra-dense lipid region with a high degree of lipid packing that excludes a larger variety of probe lipids than has been previously observed.”

“Collectively, our findings suggest that protein condensates reduce lipid mobility (Fig. 5i) and facilitate lipid probe partitioning **by reorganizing membrane lipids**, which could be related to the previous reports suggesting that protein condensates exert a substantial compressive stress on the underlying lipids⁴, leading to a tighter lipid packing and lower membrane fluidity⁵.”

4) This framing is also relevant to Figure 5, where the authors draw conclusions from an observation that probe depletion is reduced when NaCl concentration increases, but they also show that this same treatment leads to an increase in the area fraction occupied by domains (Sup fig. 2) Together, it seems reasonable to conclude that the concentration of Ni lipids within protein domains is reduced (because the same number spread out over a larger area), allowing there to be more sites for uncoupled lipids, including lipid probes. This would be a simple explanation for the change in probe partitioning reported.

Action taken: As the reviewer states, the extent of probe exclusion is reduced with the increasing NaCl concentrations in Figure 5h. However, we think the reviewer may have been confused when examining Supplementary Figure 2 (copied below). In particular, Supplementary Figure 2 shows that the **area fraction of the protein-rich domains decreases with increasing NaCl concentration**. This is the **direct opposite of the reviewer’s statement**. As such, it seems to us that this data cannot be taken to support the idea that the decrease in probe lipid depletion (with increasing NaCl concentration) arises from the dilution of DGS-Ni-NTA lipids within a larger protein-rich domain.

To help readers to better understand our findings, we made a change in Supplementary Figure 2 (copied below), such that an arrow and the phrase ‘**Increasing NaCl concentration**’ were added to indicate that the NaCl concentration is **increasing** in the direction of the arrow.

Supplementary Figure 2. Area fraction of protein-rich regions as a function of NaCl concentration and DGS-Ni-NTA concentration. A total of 20 lipid membranes were analyzed from 2-3 independent experiments for each condition. Bars represent the average area fraction for each case. Membrane composition: 75-95 mol% DOPC, 5-25 mol% DGS-Ni-NTA. 1 μ M of his-RGG labeled with Atto 488 was used.

5) Also in Figure 5, the authors report slowed mobility of the probe. Here it would make sense to compare to an expected result in which 80% (or similar %) of lipids in the leaflet adjacent to the domain are immobilized. I would expect this would slow diffusion of a free probe to some extent, possibly enough to be consistent with the subtle slowing reported in figure 5.

Author response: We appreciate the reviewer’s comment. We understand the reviewer’s point; however, it is unfortunately very difficult to estimate the fraction of DGS-Ni-NTA lipids within the protein-rich regions. Instead of assuming the fraction of DGS-Ni-NTA lipids in the protein-rich regions, we presented the possible factors that could contribute to the reduced lipid mobility in the manuscript as follows:

“The observed differences in lipid mobility could result from differences in lipid packing but could also arise from a combination of effects including differences in lipid composition, such as enrichment of DGS-Ni-NTA lipids in protein-rich regions, frictional interactions between proteins and lipids, and modified local bending rigidity or local spontaneous curvature due to membrane-bound proteins⁶.”

Minor comments

1) In the abstract, both biological examples of liquid-like condensates given involve transmembrane proteins, where transbilayer coupling is trivial. Is there a better biological motivation that does not include transmembrane proteins?

Author response: We appreciate the reviewer's comment. Understanding of membrane-bound protein condensates is a nascent field, and condensate-driven mechanisms are likely to collaborate with mechanisms that depend upon transmembrane proteins. Therefore, it is not clear at this time which biological mechanisms may depend solely on coupling between peripheral membrane proteins. We expect that the transbilayer coupling observed here works in concert with other factors, including transmembrane proteins, to achieve robust transbilayer communication.

2) I recommend not using the word "ordering" until it is defined. In the membrane field, the word "ordering" often refers to an increase in chain order parameters, which is not explored here. I am not sure, but I think here it is used to refer to trapping of certain lipids within domains?

Action taken: We appreciate the reviewer's comment. As mentioned above, we have replaced the term 'ordering' with 'reorganization' throughout the paper to prevent confusion. Also, we presented a list of the possible mechanisms by which the reorganization of lipids could occur in the manuscript as follows (newly added text in red):

"Probe exclusion arises due to the local reorganization of lipids, which could encompass changes in lipid packing¹, lipid composition (enrichment of DGS-Ni-NTA lipids in protein-rich regions), or the degree of lipid hydration owing to the presence of proteins², each via protein-lipid interactions."

3) Conventionally, the word "coupling" means that there is a cross-term in the free energy that involves thermodynamic parameters associated with each leaflet of the bilayer. In the text, this word is used regularly to describe times when domains come into registration, which is a consequence of a certain type of coupling but not coupling itself. I recommend changing how this word is used throughout to adhere to its conventional definition. Alternatively, the word should be re-defined somewhere early in the text.

Action taken: We appreciate the reviewer's comment. We have added the following phrase in the Introduction of our revised manuscript to define the transmembrane coupling in our paper (newly added text in red):

"Interestingly, we observe that phase separation of model proteins is highly coupled across the membrane surface, such that protein-enriched regions on one side of the membrane tightly colocalized with protein-enriched regions on the opposite side of the

membrane. Throughout the paper, we will refer to this transmembrane colocalization of protein-rich domains as transmembrane coupling.”

4) For Sup Fig 8, I am guessing that C/D represent different conditions of some sort? Only one condition is stated in the figure caption.

Action taken: The same membrane composition and the buffer were used for all the panels of Supplementary Figure 8. We have updated the caption for Supplementary Figure 8 as follows (newly added text in red):

“The same membrane composition (80 mol% DOPC, 15 mol% DOPS, 5 mol% PI(4,5)P2, and 0.5 mol% BODIPY TR Ceramide) and buffer (25 mM HEPES, 100 mM NaCl, pH 7.4) were used for all the panels.”

Reviewer #3:

The authors have returned a strengthened manuscript that addresses my comments, and many of the issues raised by all three reviewers. I have one minor comment that must be addressed before I'd be comfortable.

1) I am still troubled by the presentation of Figure 2 and the data on salt concentration, specifically:

– lines 166-168: “... when the transition temperature is reached, the two phases have the same concentration and therefore are indistinguishable from one another, such that one homogeneous phase exists.”. The only location where two phases become indistinguishable from one another is the critical point. Generally, for a first-order transition, there is a miscibility gap between the two phases at the transition point. I suggest that the authors remove this sentence (as it adds little to the text and is incorrect) or correct it.

Action taken: We appreciate the reviewer’s careful reading of our manuscript. As the reviewer suggested, we removed the following sentence in the revised manuscript:

“Ultimately, when the transition temperature is reached, the two phases have the same concentration and therefore become indistinguishable from one another, such that one homogenous phase exists.”

References

1. Yim, H. *et al.* Rearrangement of lipid ordered phases upon protein adsorption due to multiple site binding. *Phys. Rev. Lett.* **96**, 198101 (2006).
2. Mangiarotti, A. & Bagatolli, L. A. Impact of macromolecular crowding on the mesomorphic behavior of lipid self-assemblies. *Biochim. Biophys. Acta - Biomembr.* **1863**, 183728 (2021).
3. Bordovsky, S. S., Wong, C. S., Bachand, G. D., Stachowiak, J. C. & Sasaki, D. Y. Engineering Lipid Structure for Recognition of the Liquid Ordered Membrane Phase. *Langmuir* **32**, 12527–12533 (2016).
4. Yuan, F. *et al.* Membrane bending by protein phase separation. *Proc. Natl. Acad. Sci. U. S. A.* **118**, e2017435118 (2021).
5. Mangiarotti, A., Siri, M., Zhao, Z., Malacrida, L. & Dimova, R. Wetting by biomolecular condensates increases membrane lipid packing and dehydration. *bioRxiv* (2023).
6. Zhang, L. & Granick, S. Slaved diffusion in phospholipid bilayers. *Proc. Natl. Acad. Sci. U. S. A.* **102**, 9118–9121 (2005).

REVIEWER COMMENTS

Reviewer #2 (Remarks to the Author):

I appreciate that the authors have shifted the focus somewhat away from lipids and lipid ordering being a contributor to the observed behavior, but I do not think they have gone far enough, which I find particularly frustrating at this stage in the review process.

In my view, the data presented nicely demonstrates that protein domains assembled on different leaflets have lower free energy when in registration, but nothing to suggest that nontrivial actions by lipids are contributing to this behavior, at least in the context interrogated. Unfortunately, this is still not the story told by the authors. For example, the authors state in the abstract "How do liquid-like protein phases communicate across the lipid bilayer? Our results, based on lipid probe partitioning and the differential mobility of proteins and lipids, collectively suggest that protein condensates locally reorganize membrane lipids, which could be explained by multiple effects." While I agree that the data supports that "protein condensates locally reorganize membrane lipids," I do not think that evidence is presented to suggest that this in any way answers the question "how do protein phases communicate across the lipid bilayer." In fact, as stated in past reviews (and again below), I think the data presented no not support this hypothesis.

Re point 1: I appreciate the changes, but, maybe because point 2 was not understood, I don't think the statement "One possible explanation for these two findings is that the protein-rich region becomes enriched to some degree in DGS-NiNTA lipids." I think it would be more accurate to say "A likely explanation for these findings is that protein-rich regions are highly enriched in DGS-NiNTA lipids." I don't understand the resistance to this given the observation that the area fraction of domains tracks very strongly with the mole fraction of Ni lipids and Ni lipids interact strongly with proteins that contain a his tag, and the brightness of protein away from domains is low. All lines of evidence point to a very high concentration of Ni lipids within protein domains.

Re: point 2: I am baffled that the authors do not see the argument that $K_p = 2$ is expected in a simple limit of complete exclusion in only one leaflet – the added +1 in the numerator and denominator comes from the concentration of probe in the opposing leaflet, which in this limit is uniform. In this case, the intensity of the domain is half that of the intensity outside of the domain (since only 1 layer contains probe). There are corrections to this if domains occupy a larger area fraction when probe particle number is conserved. A detailed explanation with some equations is included as pdf.

$K_p = 2$ is the number that is observed experimentally. This simple limit seems like a reasonable explanation for observations with low Ni lipid concentration and low domain area fraction, or at least this limit should be the null hypothesis to be testing more complicated models against. Since there are good reasons to think that the domains contain a high concentration of unlabeled Ni lipids, then the probe partitioning is at largely explained by crowding effects alone.

Moreover, this simple explanation seems nicely consistent with the data presented over many figures (lipid probes all behave the same, partitioning to first order independent of domain area fraction, lipid probe contrast is roughly 0, 50%, 100% when regions are in registration, partial registration, or with no domains, probe mobility being intermediate when domains are not in registration compared to registered domains or bare membranes, etc), up to some possible minor corrections. This explanation suggests that protein condensates form on each layer independently, and that lipid organization need not contribute to any coupling mechanism between layers. This is exactly the opposite of the mechanism implied in the abstract.

Re: point 3 – I appreciate some text being removed, but It think text throughout the manuscript (not just in this section) should be updated as well.

Re: point 4: I agree that the inverted axes were confusing and the clarification is appreciated. Probes are more uniform at 200mM NaCl than 100mM NaCl for 15 mol% DGS-Ni-NTA, and the area fraction at 200mM is smaller than that at 100mM (though there is a big spread). The larger point is that the text states that changing salt "increases interactions" – but the reality is that this has many consequences, changing domain size, composition, and other physical properties of the condensate as well – The simple argument made is misleading. I think it is reasonable to say that seeing changes in various observables mean the (complicated thermodynamic) system is coupled, but does not imply a specific mechanism.

Lets imagine that the lipid organization in each leaflet can be treated independently. In this case the partition coefficient, as defined by the authors, is brightness (a proxy for probe concentration) outside divided by brightness inside. If each leaflet is considered independently, then concentration of probe in each context simply adds

$$Kp = \frac{[probe]_{outside}}{[probe]_{inside}} = \frac{[probe]_{outside,leaflet1} + [probe]_{outside,leaflet2}}{[probe]_{inside,leaflet1} + [probe]_{inside,leaflet2}}$$

For simplicity, it is easier to think about probe fractional concentration, which we can do by just multiplying by 1:

$$Kp = \frac{[probe]_{outside,leaflet1} + [probe]_{outside,leaflet2}}{[probe]_{inside,leaflet1} + [probe]_{inside,leaflet2}} \frac{\langle [probe]_{leaflet} \rangle}{\langle [probe]_{leaflet} \rangle}$$

Making the assumption that the average probe concentration is the same in both leaflets, then probe concentrations can just be replaced with the local concentration normalized by the average concentration:

$$Kp = \frac{F_{outside,leaflet1} + F_{outside,leaflet2}}{F_{inside,leaflet1} + F_{inside,leaflet2}}$$

Lets imagine that in the domain is on one leaflet, and the probe is completely excluded from the domain in in that leaflet ($F_{inside,leaflet1} = 0$) but does not impact the probe concentration at all in the over leaflet ($F_{inside,leaflet2} = 1$). In the limit of the domains taking up a small area fraction, even if probe particle number is conserved, then $F_{outside,leaflet1} = 1$ and everything becomes:

$$Kp = \frac{F_{outside,leaflet1} + F_{outside,leaflet2}}{F_{inside,leaflet1} + F_{inside,leaflet2}} = \frac{1+1}{0+1} = 2$$

Which is the number that is observed experimentally. This seems like a reasonable explanation for observations with low NTA lipid concentration and low domain area fraction.

When the area fraction of domains is not small, then it matters if probe particle number is conserved or not. If it is not conserved (which was my initial assumption), then there isn't any dependence on domain area fraction. This seems like a possibility since the pore spanning bilayers are connected to a lipid reservoir. If it is conserved, then there is a geometrical correction, because probes excluded from the domains lead to an increased concentration of probes outside of the domains. I think this is the where the authors argument regarding Kp not changing with area fraction comes from? By my math, for 20% domain coverage, then Kp becomes $(1.25+1)/(0+1)=2.25$. For 50% it becomes $(2+1)/(0+1) = 3$. Of course, a raising of $F_{inside,leaflet1}$ above 0 could offset this increase. This also seems like a reasonable thing to be happening as the area fraction of domains increases somewhat faster than the concentration of NTA lipids at 100mM NaCl.

Response to Reviews for the submission NCOMMS-23-09926B

Title

Transmembrane coupling of liquid-like protein condensates

Authors

Yohan Lee, Sujin Park, Feng Yuan, Carl C. Hayden, Liping Wang, Eileen M. Lafer, Siyoung Q. Choi and Jeanne C. Stachowiak

We appreciate the commitment that the referees have shown throughout the review process. Their feedback has helped us to see our data from new perspectives that will influence our future work.

It is important to note that all 3 referees already agree on the major claims of our paper, which are:

(i) Transbilayer coupling of liquid-like protein condensates is a novel observation that is clearly demonstrated in our experiments and is likely to be of interest to a broad community of scientists.

(ii) This coupling arises from entropy maximization within the lipid-protein system, which is the major hypothesis of the paper.

Referees 1 and 3 have already stated that they are fully satisfied with the manuscript after revision.

Referee 2 has stated in previous rounds of revision that they are satisfied with all points originally raised, except for the interpretation of data in Figure 4 on the changes in probe lipid partitioning that result from protein condensation. While we agree with the referee that it is important to provide the clearest possible presentation of these results, the debate about their interpretation does not alter the major claims of the paper, as listed above.

Below we respond to each of the Referee 2's comments and describe the further actions we have taken to address them. The changes we have made to the text are highlighted in the revised manuscript.

In summary, we find referee 2's hypotheses about our data very interesting and we have taken them very seriously. However, we believe that it remains difficult to distinguish between several possible explanations for the lipid probe partitioning. Specifically, we identify key pieces of data that appear inconsistent with the idea proposed by the referee - that probe lipid exclusion is driven predominantly by enrichment of Ni-NTA lipids in the protein-enriched phase, and cannot be explained by other mechanisms such as local ordering of lipids or suppression of lipid fluctuations, which were mentioned by referees 1 and 3. Therefore, we have chosen to highlight multiple possible explanations for the lipid probe exclusion, rather than singling out one dominant mechanism. As we describe below, we believe this approach will maximize the clarity of paper by avoiding complex and speculative arguments, which, while interesting, remain to be fully validated. It is not our intention to be stubborn or to put forward our own specific hypothesis about the

meaning of the results. Instead, we wish to leave the question of the specific mechanism open, allowing readers to draw their own conclusions and inviting future work that will refine understanding.

Reviewer Comments

Reviewer #2:

I appreciate that the authors have shifted the focus somewhat away from lipids and lipid ordering being a contributor to the observed behavior, but I do not think they have gone far enough, which I find particularly frustrating at this stage in the review process.

In my view, the data presented nicely demonstrates that protein domains assembled on different leaflets have lower free energy when in registration, but nothing to suggest that nontrivial actions by lipids are contributing to this behavior, at least in the context interrogated. Unfortunately, this is still not the story told by the authors.

For example, the authors state in the abstract “How do liquid-like protein phases communicate across the lipid bilayer? Our results, based on lipid probe partitioning and the differential mobility of proteins and lipids, collectively suggest that protein condensates locally reorganize membrane lipids, which could be explained by multiple effects.” While I agree that the data supports that “protein condensates locally reorganize membrane lipids,” I do not think that evidence is presented to suggest that this in any way answers the question “how do protein phases communicate across the lipid bilayer.” In fact, as stated in past reviews (and again below), I think the data presented do not support this hypothesis.

Author response: We are having trouble understanding the resistance of the referee to this nonspecific language in the abstract of the paper. The proteins used in our system do not span the membrane, yet they become colocalized across the membrane. Therefore, they must be reorganizing the membrane in some way or another to achieve this coupling. That is all we are saying here, and the referee seems to agree with it. There is no specific mechanism implied by this wording. In fact, we chose the words “could be explained by multiple effects” to deliberately avoid implying a specific mechanism, according to the referee’s previous requests. We do not know how to further edit this text to make it clearer that no specific mechanism has been identified.

1) I appreciate the changes, but, maybe because point 2 was not understood, I don’t think the statement “One possible explanation for these two findings is that the protein-rich region becomes enriched to some degree in DGS-NiNTA lipids.” I think it would be more accurate to say “A likely explanation for these findings is that protein-rich regions are highly enriched in DGS-Ni-NTA lipids.” I don’t understand the resistance to this given the observation that the area fraction of domains tracks very strongly with the mole fraction of Ni lipids and Ni lipids interact strongly with proteins that contain a his tag, and the brightness of protein away from domains is low. All lines of evidence point to a very high concentration of Ni lipids within protein domains.

Author response: The referee continues to propose that displacement of the probe lipids by enrichment of the DGS-NiNTA lipids (to which the protein binds) is the single most likely explanation for the exclusion of probe lipids from the protein-enriched regions. We found this

hypothesis interesting and have taken it very seriously over the last two rounds of review, collecting and analyzing a significant amount of new data. In our interpretation, the resulting data does not support this hypothesis. It is for this reason that we have chosen to deliberately avoid highlighting one particular mechanism of probe exclusion as more likely than the others. Instead, we have listed multiple mechanisms and left it to future work to determine which, if any, of the possible mechanisms plays a dominant role over the others.

We would like to try to explain more clearly why we are not fully convinced by the hypothesis that the probe lipids are displaced by enrichment of Ni-NTA lipids. Having read again over our correspondence with this referee, we think that perhaps we have not communicated clearly enough about the difference in size between the proteins and lipids, and how this size difference impacts our expectations about the likely extent of enrichment of Ni-NTA lipids.

Specifically, as noted in the revised manuscript, the radius of gyration (R_g) of the FUS LC protein is approximately 3 nm¹. We would expect the RGG protein to have a similar radius of gyration, owing to its similar molecular weight and the fact that both proteins are disordered. Based on these approximations, the projected area of RGG on a two-dimensional surface should be approximately 30 nm² (πR_g^2). Now, given that the area of the lipid headgroup is approximately 0.7 nm², there would be around 40 lipids beneath a single membrane-bound RGG protein (30 nm²/0.7 nm² ~40). Each protein binds to the membrane using 6x histidine tag. However, due to steric hindrance created by binding of histidine to Ni-NTA, the number of Ni-NTA lipids bound to each protein is unlikely to be more than two to three². Therefore, once the fraction of Ni-NTA lipids in the membrane reaches 5-7.5 mol% (2/40 = 0.05; 3/40 = 0.075), there is sufficient Ni-NTA lipid available to fully saturate the membrane surface with proteins, as is likely the case within the protein-enriched phase. The implication of this analysis is that, once the concentration of Ni-NTA lipids in the membrane significantly exceeds 5-7.5 mol%, protein condensation should provide little driving force for further accumulation of Ni-NTA lipids in the membrane, as the histidine tags within the protein condensate are already saturated by Ni-NTA lipids. Even in the unlikely case that all 6 histidine residues bind to Ni-NTA lipids, a Ni-NTA lipid concentration of 15 mol% should be enough to fully saturate the binding capacity of the protein condensate.

Based on this logic, we designed the experiment in Supplementary Figure 7 (copied below) to address the referee's hypothesis. Here we examined the partition coefficient of the probe dye, Texas Red DHPE, as a function of the Ni-NTA lipid concentration, which we examined at 5 mol%, 15 mol%, and 25 mol%. The hypothesis of the referee is that probe exclusion results from a local increase in the concentration of Ni-NTA lipids within the protein-rich phase. If this is the case, we should observe a substantial drop in the partition coefficient of the dye with increasing concentration of Ni-NTA lipids. In particular, once the concentration significantly exceeds the range identified above (5-15 mol%), assembly of the protein-rich phase should cease to require a local increase in Ni-NTA lipid concentration, such that the dye partition coefficient should approach a value of one.

In contrast, the data in Supplementary Figure 7 show that the partition coefficient of the dye changes very little as the concentration of Ni-NTA lipids increases. In our interpretation, these

data are inconsistent with the referee's hypothesis and therefore suggest that exclusion of probe lipids is unlikely to arise primarily from local enrichment of Ni-NTA lipids.

Nonetheless, we fully acknowledge that the mechanism of dye exclusion is not entirely clear at this stage, and may arise from multiple effects. For this reason, we have acknowledged multiple possible mechanisms, including the hypothesis proposed by the referee, in the text.

We presented this data in Supplementary Figure 7 in our first response. However we did not clearly articulate the size difference between the proteins and lipids at that time. Therefore, the referee may have been confused by the data. If that is the case, we apologize and thank the referee once again for their time.

Action Taken: We feel it is best not to add the arguments above to the manuscript, as they remain speculative. However, to clarify that no definitive mechanism of dye exclusion has been identified at this stage, we added the following sentence to the revised manuscript (newly added text in red):

“Why are diverse probe lipids excluded from the protein-enriched phase? Probe exclusion arises due to the local reorganization of lipids, which could encompass changes in lipid packing³, lipid composition (enrichment of DGS-Ni-NTA lipids in protein-rich regions), or the degree of lipid hydration owing to the presence of proteins⁴, each via protein-lipid interactions. To quantify the extent of probe lipid exclusion, we defined a partition coefficient (K_P) as $K_P = I_B / I_D$, where I_B and I_D indicate the fluorescence intensity of the brighter and the dimmer regions in the lipid channel after subtracting the background intensity, respectively⁵. Based on the definition of K_P , the more the probes are excluded from the dimmer region, the higher the K_P value will become.

Interestingly, the extent of probe lipid (Texas Red-DHPE) exclusion remained constant across a range of DGS-Ni-NTA concentrations (5, 15, 25 mol%) (Supplementary Figs. 6-7). Meanwhile, the area fraction of protein-rich regions increased over the same range of DGS-Ni-NTA concentrations (Supplementary Fig. 2). One possible explanation for these two findings is that the protein-rich region becomes enriched to some degree in DGS-Ni-NTA lipids. Once a stable degree of enrichment is achieved within these regions, increasing the DGS-Ni-NTA concentration serves only to increase the area fraction of the protein-rich phase. However, our data at present cannot distinguish between several possible mechanisms for probe lipid exclusion, listed above.”

Supplementary Figure 7. Partition coefficients (K_p) of lipid probe as a function of different DGS-Ni-NTA concentrations in the membrane. Red bars indicate the average K_p values for each case. Membrane composition: 75-95 mol% DOPC, 5-25 mol% DGS-Ni-NTA with 0.5 mol% Texas Red DHPE. 1 μ M of his-RGG was used. Brackets show statistically significant comparisons using an unpaired, two-tailed Student's t test. (n.s. indicates a difference that was not statistically significant.)

2) I am baffled that the authors do not see the argument that $K_p = 2$ is expected in a simple limit of complete exclusion in only one leaflet – the added +1 in the numerator and denominator comes from the concentration of probe in the opposing leaflet, which in this limit is uniform. In this case, the intensity of the domain is half that of the intensity outside of the domain (since only 1 layer contains a probe). There are corrections to this if domains occupy a larger area fraction when probe particle number is conserved. A detailed explanation with some equations is included as pdf.

$K_p = 2$ is the number that is observed experimentally. This simple limit seems like a reasonable explanation for observations with low Ni lipid concentration and low domain area fraction, or at least this limit should be the null hypothesis to be testing more complicated models against. Since there are good reasons to think that the domains contain a high concentration of unlabeled Ni lipids, then the probe partitioning is largely explained by crowding effects alone.

Moreover, this simple explanation seems nicely consistent with the data presented over many figures (lipid probes all behave the same, partitioning to first order independent of domain area fraction, lipid probe contrast is roughly 0, 50%, 100% when regions are in registration, partial registration, or with no domains, probe mobility being intermediate when domains are not in registration compared to registered domains or bare membranes, etc), up to some possible minor

corrections. This explanation suggests that protein condensates form on each layer independently, and that lipid organization need not contribute to any coupling mechanism between layers. This is exactly the opposite of the mechanism implied in the abstract.

Author response: We appreciate the reviewer taking the time to think carefully about our data. Thanks to the reviewer's detailed explanation, we now understand how the reviewer calculated the partition coefficient (K_P) of the probe lipid. We acknowledge that K_P should have a value between 1 and 2 if (i) the area fraction of the protein-rich domains is **very small**, and (ii) the lipid composition of each leaflet is entirely independent of the other leaflet. However, as we will explain, these assumptions are inconsistent with our data. Further, the data in Supplementary Figure 7 (copied above), demonstrate that partition coefficients above 2 are frequently observed in our system.

First, as shown by the data in Supplementary Figure 2 (copied below), the area fraction of the protein-rich phase is not small in our experiments. It approaches 100% in several of our experimental conditions. Therefore, by the referee's arguments, we would expect to see an increase in partition coefficient with increasing area fraction, as the probe lipids displaced from the protein-enriched regions would begin to accumulate in the surrounding protein-dilute phase. This increase is not observed, as shown in Supplementary Figure 7.

Second, the lipid composition of the two leaflets is unlikely to be uncoupled, as the referee assumes. As already noted, this assumption would cap the partition coefficient at a value of 2, while our data include many values significantly above this threshold (Supplementary Figure 7). Further, coupling between the lipids, though the mechanism has not been fully elucidated by our work, must be occurring in order for the protein-rich phases to become coupled across the membrane, as they consist of peripheral membrane proteins that can only interact through the lipid medium.

Lastly, it is important to note that the referee's idea that the partition coefficient should fall between 1 and 2, even if correct, would not imply any particular mechanism of coupling, as the assumptions on which the estimate is based (uncoupled leaflets in the limit of a small protein-enriched phase) could be applied equally to any of the proposed mechanisms: Ni-NTA lipid enrichment, local ordering/compression of lipids, local suppression of lipid fluctuations, etc.

Supplementary Figure 2. Area fraction of protein-rich regions as a function of NaCl concentration and DGS-Ni-NTA concentration. A total of 20 lipid membranes were analyzed from 2-3 independent experiments for each condition. Bars represent the average area fraction for each case. Membrane composition: 75-95 mol% DOPC, 5-25 mol% DGS-Ni-NTA. 1 μ M of his-RGG labeled with Atto 488 was used.

3) I appreciate some text being removed, but I think text throughout the manuscript (not just in this section) should be updated as well.

Action taken: This comment was addressed by the text change in response to the second comment above.

4) I agree that the inverted axes were confusing and the clarification is appreciated. Probes are more uniform at 200mM NaCl than 100mM NaCl for 15 mol% DGS-Ni-NTA, and the area fraction at 200mM is smaller than that at 100mM (though there is a big spread). The larger point is that the text states that changing salt “increases interactions” – but the reality is that this has many consequences, changing domain size, composition, and other physical properties of the condensate as well – The simple argument made is misleading. I think it is reasonable to say that seeing changes in various observables means the (complicated thermodynamic) system is coupled, but does not imply a specific mechanism.

Action taken: As the reviewer pointed out, we acknowledge that changing the NaCl concentration could affect multiple variables, including the domain size, lipid composition, etc. We have changed the description of Figure 5h (copied below) to better reflect this point (newly added text in red):

“We observed that decreasing the concentration of NaCl, which is expected to strengthen interactions among RGG domains, resulted in increasing values of K_P (Fig. 5h). This suggests that strengthening protein-protein interactions results in greater depletion of probe lipids from the underlying lipid bilayer. However, it is important to note that the change in NaCl concentration also impacts the area fraction of the protein-enriched phase (Supplementary Fig. 2), and may alter its lipid composition.”

Figure 5h. Transmembrane coupling of protein condensates correlates with changes in lipid organization and mobility. **h**, Left: Representative microscopic image in the lipid channel showing both brighter and dimmer regions for calculating the partition coefficient of the lipid probe. Right: Partition coefficients (K_P) as a function of different NaCl concentrations. The red bars indicate the average K_P values for each case. Membrane composition: 85 mol% DOPC, 15 mol% DGS-Ni-NTA with 0.5 mol% Texas Red-DHPE. 1 μM of unlabeled his-RGG was used. Scale bars, 5 μm .

References

1. Perdikari, T. M. *et al.* A predictive coarse-grained model for position-specific effects of post-translational modifications. *Biophys. J.* **120**, 1187–1197 (2021).
2. Nye, J. A. & Groves, J. T. Kinetic control of histidine-tagged protein surface density on supported lipid bilayers. *Langmuir* **24**, 4145–4149 (2008).
3. Yim, H. *et al.* Rearrangement of lipid ordered phases upon protein adsorption due to multiple site binding. *Phys. Rev. Lett.* **96**, 198101 (2006).
4. Mangiarotti, A. & Bagatolli, L. A. Impact of macromolecular crowding on the mesomorphic behavior of lipid self-assemblies. *Biochim. Biophys. Acta - Biomembr.* **1863**, 183728 (2021).
5. Bordovsky, S. S., Wong, C. S., Bachand, G. D., Stachowiak, J. C. & Sasaki, D. Y. Engineering Lipid Structure for Recognition of the Liquid Ordered Membrane Phase. *Langmuir* **32**, 12527–12533 (2016).